# A hardware system for real-time decoding of in vivo calcium imaging data

Zhe Chen[1,2], Garrett J Blair[2], Changliang Guo[3,4], Jim Zhou[1], Juan-Luis Romero-Sosa[2], Alicia Izquierdo[2,5], Peyman Golshani[3,4,5], Jason Cong[1], Daniel Aharoni[3,4,5], Hugh T Blair[2,5]*

[1]Department of Electrical and Computer Engineering, University of California, Los Angeles, Los Angeles, United States; [2]Department of Psychology, University of California, Los Angeles, Los Angeles, United States; [3]David Geffen School of Medicine, University of California, Los Angeles, Los Angeles, United States; [4]Department of Neurology, David Geffen School of Medicine, University of California, Los Angeles, Los Angeles, United States; [5]Integrative Center for Learning and Memory, University of California, Los Angeles, Los Angeles, United States

*For correspondence:
tadblair@g.ucla.edu

**Abstract** Epifluorescence miniature microscopes ('miniscopes') are widely used for in vivo calcium imaging of neural population activity. Imaging data are typically collected during a behavioral task and stored for later offline analysis, but emerging techniques for online imaging can support novel closed-loop experiments in which neural population activity is decoded in real time to trigger neurostimulation or sensory feedback. To achieve short feedback latencies, online imaging systems must be optimally designed to maximize computational speed and efficiency while minimizing errors in population decoding. Here we introduce *DeCalciOn*, an open-source device for real-time imaging and population decoding of in vivo calcium signals that is hardware compatible with all miniscopes that use the UCLA Data Acquisition (DAQ) interface. DeCalciOn performs online motion stabilization, neural enhancement, calcium trace extraction, and decoding of up to 1024 traces per frame at latencies of <50 ms after fluorescence photons arrive at the miniscope image sensor. We show that DeCalciOn can accurately decode the position of rats ($n$ = 12) running on a linear track from calcium fluorescence in the hippocampal CA1 layer, and can categorically classify behaviors performed by rats ($n$ = 2) during an instrumental task from calcium fluorescence in orbitofrontal cortex. DeCalciOn achieves high decoding accuracy at short latencies using innovations such as field-programmable gate array hardware for real-time image processing and contour-free methods to efficiently extract calcium traces from sensor images. In summary, our system offers an affordable plug-and-play solution for real-time calcium imaging experiments in behaving animals.

## Editor's evaluation

This article presents a novel realtime processing pipeline for miniscope imaging which enables accurate decoding of behavioral variables and generation of feedback commands within less than 50ms. The efficiency of this important tool for experiments requiring close-loop interaction with brain activity is demonstrated based on compelling measurements in two experimental contexts. This pipeline will be useful for a wide range of questions in system neuroscience.

## Introduction

Miniature epifluorescence microscopes ('miniscopes') can be worn on the head of an unrestrained animal to perform in vivo calcium imaging of neural population activity during free behavior (*Ghosh*

*et al., 2011*; *Ziv et al., 2013*; *Cai et al., 2016*; *Aharoni et al., 2019*; *Hart et al., 2020*). Imaging data are usually collected while subjects are engaged in a task and stored for later offline analysis. Popular offline analysis packages such as CalmAn (*Giovannucci et al., 2019*) and MIN1PIPE (*Deng et al., 2015*) employ algorithms (*Pnevmatikakis et al., 2016*) for demixing crossover fluorescence between multiple sources to extract calcium traces from single neurons, but these algorithms cannot be implemented in real time because they incur significant computing delays and rely on acausal computations. Emerging techniques for online trace extraction (*Friedrich et al., 2017*; *Mitani and Komiyama, 2018*; *Chen et al., 2019*; *Chen et al., 2020*; *Chen et al., 2022a*; *Friedrich et al., 2021*; *Taniguchi et al., 2021*) offer potential for carrying out real-time imaging experiments in which closed-loop neurostimulation or sensory feedback are triggered at short latencies in response to neural activity decoded from calcium fluorescence (*Aharoni and Hoogland, 2019*; *Zhang et al., 2018*; *Liu et al., 2021*). Such experiments could open new avenues for investigating the neural basis of behavior, developing brain–machine interface devices, and preclinical testing of neurofeedback-based therapies for neurological disorders (*Grosenick et al., 2015*). To advance these novel lines of research, it is necessary to develop and disseminate accessible tools for online calcium imaging and neural population decoding.

Here we introduce *DeCalciOn*, a plug-and-play hardware device for *De*coding *Calci*um Images *On*line that is compatible with existing miniscope devices which utilize the UCLA Miniscope DAQ interface board (*Cai et al., 2016*; *Scott et al., 2018*; *de Groot et al., 2020*; *Blair et al., 2021*). We show that this system can decode hippocampal CA1 calcium activity in unrestrained rats (n = 12) running on a linear track, and can also categorically classify behaviors performed by rats (n = 2) during an instrumental touchscreen task from orbitofrontal cortex (OFC) calcium activity. Performance tests show that the system can accurately decode up to 1024 calcium traces to trigger TTL outputs within <50 ms of fluorescence detection by the miniscope's image sensor. To achieve these short real-time decoding latencies, DeCalciOn relies upon customized hardware for online image processing (*Chen et al., 2019*; *Chen et al., 2022a*) running in the fabric of a field-programmable gate array (FPGA); the system also utilizes efficient software for trace decoding (*Chen et al., 2022b*) that achieves microsecond variability in decoding latencies by running on an ARM core under the FreeRTOS real-time

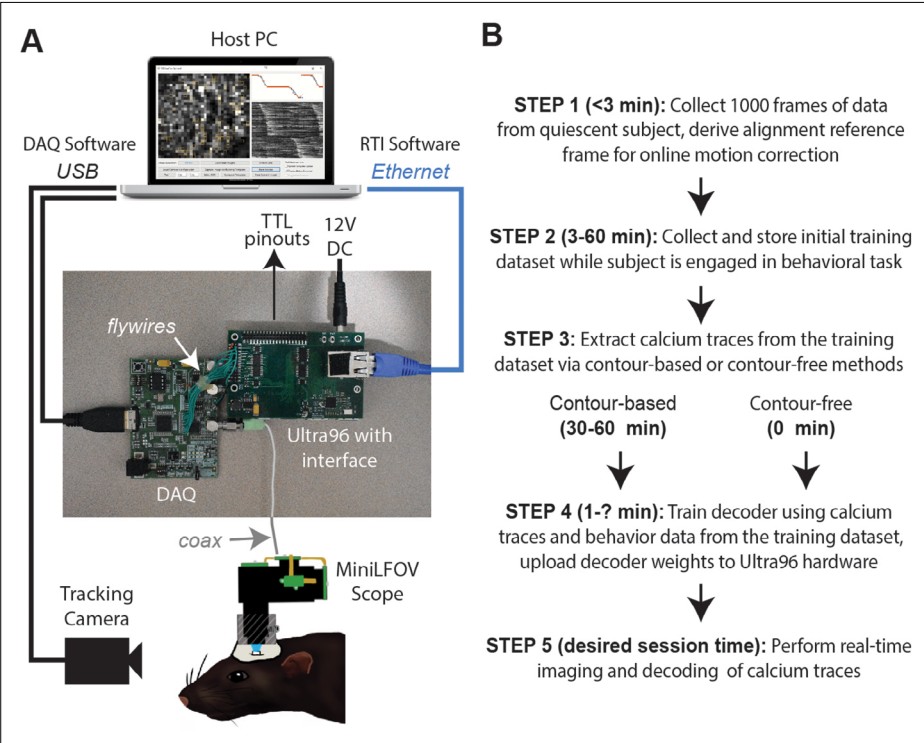

**Figure 1.** Real-time imaging protocol and system hardware. (**A**) Sequence of steps for a real-time imaging and decoding session. (**B**) Miniscope connects to DAQ via coax cable, DAQ connects to Ultra96 via flywires, host PC connects to Ultra96 via Ethernet and to DAQ via USB 3.0; TTL pinouts from Ultra96 can drive closed-loop feedback stimulation from external devices.

operating system. We show that the time required for training the real-time decoder from example data (prior to the initiation of online imaging) can be minimized by employing contour-free methods for defining regions of interest (ROIs) from which calcium traces are extracted.

In summary, DeCalciOn provides the research community with a low-cost, open-source, easy-to-use hardware platform for short latency real-time decoding of calcium trace population activity. All hardware, software, and firmware are openly available through miniscope.org.

## Results

The DeCalciOn hardware system is implemented on an Avnet Ultra96 development board featuring a Xilinx Zynq Ultrascale + multiprocessor system-on-a-chip (MPSoC) with 2-GB DRAM (*Figure 1A*). A custom interface board mated to the Ultra96 receives real-time image data from a modified version of the UCLA Miniscope Data Acquisition (DAQ) interface, which has 14 flywires soldered to the printed circuit board (PCB) for transmitting deserialized video to the Ultra96 (pre-modified DAQ boards can be obtained through miniscope.org). Throughout online imaging sessions, the DeCalciOn system is controlled from a host PC running standard Miniscope DAQ software (to focus the lens and adjust the LED light source) alongside newly developed real-time interface (RTI) software which communicates with our custom ACTEV (Accelerator for Calcium Trace Extraction from Video) hardware accelerator, which runs in the fabric of the MPSoC's FPGA. Raw video data (including real-time motion correction vectors) and calcium trace values are transmitted via Ethernet from the Ultra96 to a host PC for storage, so that additional analyses of calcium data can be performed offline. ACTEV accelerator for the two versions of the UCLA Miniscope used here (MiniLFOV and V4, both sampling at a frame rate of ~20 Hz), as well as RTI software resources for the host PC, can be found at https://github.com/zhe-ch/ACTEV, (*Chen, 2023* copy archived at swh:1:rev:aa6393d3bd2dd490aa5369e1f2677e85e8a64a82).

### Steps of a real-time imaging session

DeCalcion is designed for conducting real-time calcium imaging experiments in freely behaving animals. Although the animal species, targeted brain regions, and behavioral tasks will necessarily differ depending on the goals of a given experiment, real-time imaging sessions conducted with DeCalciOn will generally be carried out by performing a sequence of five common steps (*Figure 1B*).

Step 1 is to mount the miniscope on the subject's head and collect 1000 frames of data to derive a reference alignment image for online motion correction. It is preferable for the subject to be in a quiescent state during this initial data collection period, to minimize brain motion while the reference image data are being stored. Imaging and behavioral data are transmitted via Ethernet from the Ultra96 for storage on the host PC. After 1000 frames of data have been stored to the host PC (~50 s after the start of acquisition at the ~20 Hz frame rates used here), imaging acquisition automatically pauses and the user is prompted to select a 128 × 128 pixel subregion of the image to use for motion correction. The host PC derives a reference image for motion correction by computing the mean image within the selected area over the 1000 frames of stored data. The reference image is then uploaded to the Ultra96 and data collection resumes with online motion correction enabled. Collecting the alignment data and uploading the reference frame takes ~3 min.

Step 2 is to collect an initial set of motion-corrected data for training the decoder. Online motion correction vectors are derived in real time for each frame and uploaded to the host PC for storage along with imaging and behavioral data. The time required to collect initial training data depends upon how much data is needed to train the decoder, which in turn depends upon the quality of calcium signals and what behaviors the decoder will be trained to predict. In performance tests presented here (see below), asymptotic accuracy for decoding a rat's position on a linear track was achieved with as little as ~3 min of training data whereas asymptotic accuracy for classifying categorical behaviors during an instrumental task required 20–40 min of training data (~100 experimental trials). The size of the training dataset required for a given experiment can be determined in advance (prior to real-time imaging sessions) by storing offline imaging data from the brain ROI while subjects are engaged in the experimental task, then carrying out offline simulations of real-time decoder performance with training sets of varying size to determine the minimum amount of training data needed for accurate decoding.

Step 3 is to pause data acquisition so that the host PC can perform analyses to extract calcium traces from each frame of the initial training dataset. Calcium traces can be extracted by two methods: a contour-based method that requires 30–60 min of processing time on the host PC during Step 3 (for reasons explained below), or a contour-free method that requires no processing time during Step 3 because this method allows calcium traces to be extracted online and stored to the host PC concurrently with collection of the initial training dataset, thereby obviating the need to devote additional time or effort to calcium trace extraction during Step 3. It is preferable for the miniscope to remain attached to the animal's head throughout Step 3 (and Step 4), because removing and remounting the miniscope can introduce shifts in the focus or alignment of the brain image and thereby degrade the accuracy of subsequent real-time decoding. Since the contour-based method of calcium trace extraction consumes considerable processing time to perform offline identification of cell contours, it also incurs the inconvenience of monitoring the subject during this extended time period while the miniscope remains attached to the head.

Step 4 is to use extracted calcium traces for training a decoder to predict behavior from vectors of calcium trace values. The time required for this depends upon the complexity of the user-defined decoder that is being trained (the decoder is trained on the host PC and runs in real time on the Ultra96 ARM core; see 'Stage 4: population decoding'). Thus, a multilayer deep network decoder would take longer to train than a single-layer linear classifier of the kind used here for performance testing. Training consumed <60 s for the linear classifiers used in performance tests presented below. Trained decoder weights (along with derived cell contour pixel masks if contour-based decoding was used) are uploaded from the host PC to the Ultra96 (a process which takes a maximum of 5 min to complete). Once the decoder weights have been uploaded, DeCalciOn is fully configured for real-time decoding of calcium traces.

Step 5 is to resume the behavior experiment while DeCalciOn performs real-time imaging and decoding of calcium traces to generate feedback output at TTL pins.

## Real-time image processing pipeline

*Figure 2* shows a schematic diagram of DeCalciOn's real-time image processing pipeline, which performs four sequential stages of image processing and calcium trace decoding: (1) online motion stabilization, (2) image enhancement via background removal, (3) calcium trace extraction, and (4) fluorescence vector decoding. As explained below, Stages 1–3 are performed by our custom ACTEV hardware accelerator (*Chen et al., 2019*; *Chen et al., 2020*; *Chen et al., 2022a*) running in the programmable logic fabric of the FPGA, whereas Stage 4 is performed by a C++ program running under the FreeRTOS operating system on the MPSoC's embedded ARM core.

### Stage 1: motion stabilization

Incoming frames arriving to the Ultra96 from the Miniscope DAQ are cropped to a 512 × 512 subregion, manually selected to contain the richest area of fluorescing neurons in each animal. Pixel data from the cropped subregion are stored to a BRAM buffer on the Ultra96 where it can be accessed by ACTEV accelerator running in the FPGA logic fabric. To correct for translational movement of brain tissue, a 128 × 128 pixel area with distinct anatomical features is manually selected during Stage 1 (*Figure 1B*) from within the 512 × 512 imaging subwindow to serve as a motion stabilization window. ACTEV's image stabilization algorithm (*Chen et al., 2019*) rigidly corrects for translational movement of brain tissue by convolving the 128 × 128 stabilization window in each frame with a 17 × 17 contrast filter kernel, and then applying a fast 2D FFT/IFFT-based algorithm to correlate the stabilization window contents with a stored reference template (derived by averaging 1000 contrast-filtered frames from the beginning of each experimental session) to obtain a 2D motion displacement vector for the current frame. *Video 1* demonstrates online performance of ACTEV's real-time motion stabilization algorithm.

### Stage 2: background removal

After motion stabilization, ACTEV removes background fluorescence from the 512 × 512 image by performing a sequence of three enhancement operations (*Chen et al., 2020*): smoothing via convolution with a 3 × 3 mean filtering kernel, estimating the frame background via erosion and dilation with a 19 × 19 structuring element (*Lu et al., 2018*), and subtracting the estimated background from the

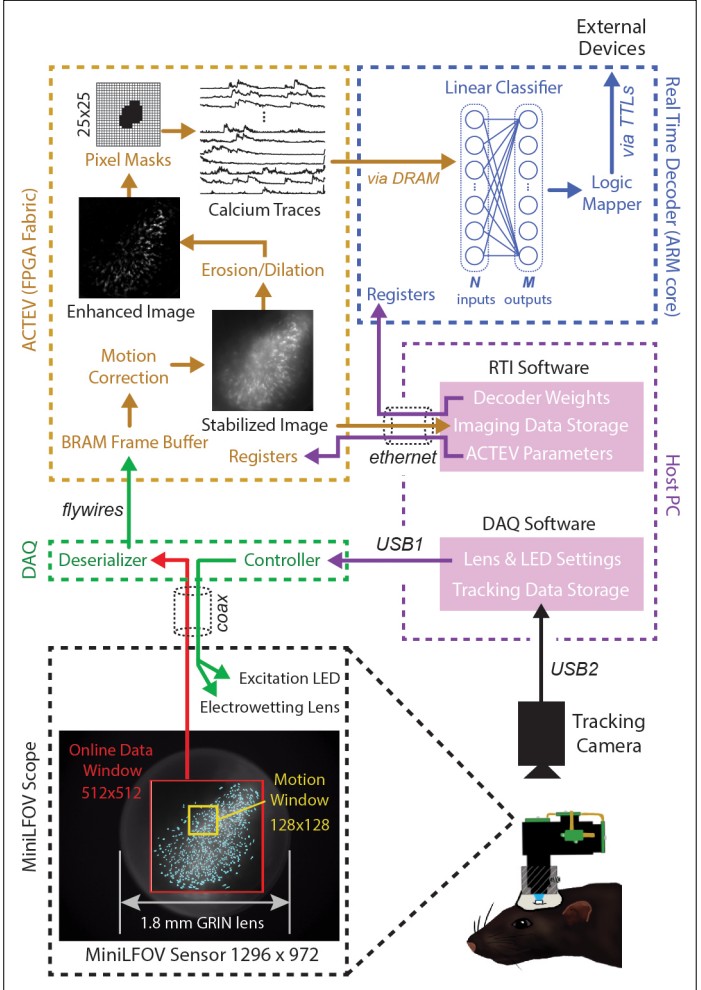

**Figure 2.** Online imaging and control pipelines. Serialized video data from the MiniLFOV are transmitted through a coaxial tether cable to the DAQ, where it is deserialized and transmitted via the flywire bus to ACTEV (Accelerator for Calcium Trace Extraction from Video) firmware programmed on the field-programmable gate array (FPGA) of the Ultra96. ACTEV crops the incoming image from its original size (1296 × 972 for the LFOV sensor shown here) down to a 512 × 512 subwindow (manually selected to contain the richest area of fluorescing neurons in each animal) before storing video frames to a BRAM buffer on the FPGA. Subsequent motion correction and calcium trace extraction steps are performed by the FPGA fabric as described in the main text. Extracted calcium traces are stored to DRAM from which they can be read as inputs to a decoder algorithm running on the Ultra96 ARM core. Decoder output is fed to a logic mapper that can trigger TTL output signals from the Ultra96, which in turn can control external devices for generating closed-loop feedback.

smoothed image. These operations produce an enhanced image in which fluorescing neurons stand out in contrast against the background (see *Video 2*; 'Enhanced image' in *Figure 2*). Calcium traces are then extracted (by methods described under 'Stage 3') from the motion-corrected (Stage 1) and background-removed (Stage 2) image.

## Stage 3: trace extraction

To derive calcium trace values, the enhanced image is filtered through a library of up to 1024 binary pixel masks, each of which defines a unique ROI for extracting a calcium trace. Each mask is defined as a selected subset of pixels within a 25 × 25 square region that can be centered anywhere in the 512 × 512 imaging subwindow (*Figure 2*, upper left). In performance test results presented below, decoding accuracy was compared for two different methods of selecting ROI pixels: *contour-based* versus *contour-free*. For contour-based ROI selection, the CaImAn (*Giovannucci et al., 2019*) algorithm was used during the intermission period to identify contours of individual neurons in the training

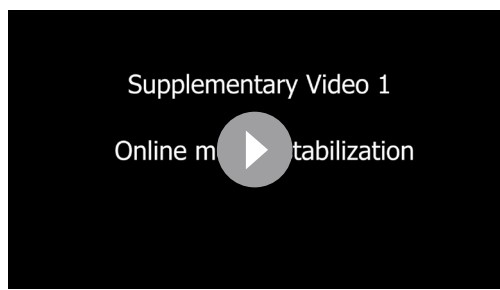

**Video 1.** Real-time motion correction. The left and right windows show sensor video data before and after motion correction, respectively. The line graphs at bottom show x (yellow) and y (green) components of the image displacement between frames before (left) and after (right) motion correction.

https://elifesciences.org/articles/78344/figures#video1

dataset, and pixel mask ROIs were then aligned to the identified regions where neurons were located. For contour-free ROI selection, the 512 × 512 imaging subwindow pixel was uniformly covered with a 32 × 32 array of square tiles, each measuring 16 × 16 pixels, which served as ROIs for calcium trace extraction. This contour-free approach did not align ROIs with individual neurons, so each extracted calcium trace could detect fluorescence originating from more than one neuron. It is shown below that this method of contour-free ROI selection can improve the computational efficiency of online decoding at little or no cost to prediction accuracy. After ROIs had been defined by the contour-based or contour-free method, calcium traces were derived by summing pixel intensities within each ROI. Demixing was not performed during real-time calcium trace extraction; it is shown below that similar prediction accuracy can be achieved with offline (demixed) versus online (non-demixed) decoding using identical contour-based ROIs.

## Stage 4: population vector decoding

The vector of calcium trace values extracted from each frame is stored to the Ultra96 DRAM, from which trace values are then read into the MPSoC's ARM core. The ARM core executes user-defined decoding algorithms running as a C++ program under the FreeRTOS operating system (which affords highly consistent execution times and therefore minimizes frame-to-frame variability in decoding latency). In performance tests reported below, we used simple linear classifier algorithms to decode calcium population vector activity of neurons imaged in CA1 and OFC of freely behaving rats. However, the C++ programmable ARM core is capable of implementing a wide range of different decoder architectures, including deep convolutional neural networks and long short-term memory networks (*Chen et al., 2022b*).

## Performance testing

Results presented below were obtained using two different versions of the UCLA Miniscope: CA1 position decoding tests were conducted using the Large Field-of-View (*Blair et al., 2021*) (MiniLFOV) version featuring 1296 × 972 pixel resolution sampled at 22.8 fps, whereas OFC instrumental behavior decoding tests were conducted using the V4 version featuring 600 × 600 pixel resolution sampled at 19.8 fps. For performance tests, we used a virtual sensor (see Methods) that was capable of feeding stored image data to the ACTEV accelerator at exactly the same frame rate (19.8 fps for V4, 22.8 fps for MiniLFOV) as raw video data arriving in real time. This allowed different real-time image processing algorithm implementations (e.g., contour-based vs. contour-free decoding) to be compared and benchmarked on the same stored datasets. Calcium traces obtained with the virtual sensor were verified to be identical with those obtained during real-time imaging sessions. ACTEV

**Video 2.** Real-time interface (RTI) view of contour-based calcium trace extraction. The online image display (left window) shows the motion corrected and enhanced (i.e., background subtracted) sensor image data as it arrives in real time from the Ultra96 board. The right window shows a scrolling display of 63 selected calcium traces from regions outlined by colored borders in the image display window. These traces are derived on the Ultra96 by summing fluorescence within their respective contour regions, and the resulting trace values are transmitted (along with sensor image data) via ethernet to the host PC for display in the RTI window. For demonstration purposes, traces shown in the window on the right are normalized within the ranges of their own individual minimum and maximum values.

https://elifesciences.org/articles/78344/figures#video2

accelerators for virtual sensor testing and for real-time (non-virtual) imaging with both versions of the UCLA Miniscope (MiniLFOV and V4) can be found at https://github.com/zhe-ch/ACTEV, (*Chen, 2023* copy archived at swh:1:rev:aa6393d3bd2dd490aa5369e1f2677e85e8a64a82).

## Position decoding from CA1 cells

For the first set of performance tests, the MiniLFOV scope was used to image neurons in the hippocampal CA1 region while Long-Evans rats (*n* = 12) ran back and forth on a 250-cm linear track (*Figure 3A*). Each linear track session was ~7 min in duration, yielding 8K–9K frames of calcium trace data per rat. CA1 pyramidal neurons are known to behave as 'place cells' that fire selectively when an animal traverses specific locations in space (*O'Keefe and Dostrovsky, 1971*), so a rodent's position can be reliably decoded from CA1 population activity (*Ziv et al., 2013*; *Cai et al., 2016*; *Wilson and McNaughton, 1993*; *Kinsky et al., 2018*; *Center for Brains Minds Machines, 2017*). As expected, many of the neurons we imaged in rat CA1 were spatially tuned on the linear track (see example tuning curves in *Figure 3C, D*; summary data in panels D, E of Supplement to *Figure 3*). For real-time decoding, we did not define a decision boundary for classifying CA1 traces as coming from 'place' versus 'non-place' cells; calcium traces were decoded from all ROIs regardless of their spatial tuning properties.

When hippocampal activity is analyzed or decoded offline (rather than online), it is common practice to perform *speed filtering* that omits time periods when the rat is sitting still. This is done because during stillness, the hippocampus enters a characteristic 'sharp wave' EEG state during which place cell activity is less reliably tuned for the animal's current location (*Buzsáki, 2015*). Here, speed filtering was not implemented during online decoding of CA1 fluorescence because the linear classifier received only real-time calcium trace data and not position tracking data that would be needed for speed filtering. As shown below, DeCalciOn achieved accurate decoding of the rat's position from calcium traces without speed filtering.

ACTEV can utilize a real-time spike inference engine to convert each frame's calcium trace value into an inferred spike count per frame. However, when decoding information from spike events (e.g., from single-unit recordings of neural population activity), it is standard practice to perform a temporal integration step by counting spikes in time bins of a specified width. In calcium imaging experiments, the GCamp molecule itself serves as a temporal integrator of spike activity. The GCamp7s indicator used in our CA1 imaging experiments has a decay constant of several hundred ms (*Dana et al., 2019*), which is similar to the width of time bins typically used for counting spikes when decoding a rat's position from single-unit recordings of place cells (*Ziv et al., 2013*; *Cai et al., 2016*; *Wilson and McNaughton, 1993*; *Kinsky et al., 2018*; *Center for Brains Minds Machines, 2017*). In accordance with prior studies of position decoding from CA1 calcium fluorescence (*Tu et al., 2020*), we found (see panel A of Supplement to *Figure 3*) that the linear classifier was more accurate at decoding the rat's position from raw calcium trace amplitudes (which chemically time-integrated spikes on a time scale of the several hundred milliseconds, corresponding to the GCaMP7s decay rate) than from inferred spike counts (which computationally time-integrated spike counts on a much shorter time scale of ~50 ms, corresponding to the miniscope's frame interval). Hence, we decoded the rat's position from raw calcium traces rather than inferred spike counts in performance tests reported below.

The linear classifier was trained on calcium trace and position data from the first half of each session, then tested on data from the second half of the session. The classifier's output vector consisted of 12 units that used a Gray coding scheme (*Figure 3B*) to represent 24 position bins (each ~20 cm wide) along a circularized representation of the track. After the linear classifier had been trained on data from the first portion of the session, learned weights were uploaded from the host PC to the Ultra96 and image data from the second half of the session (test data) was fed through the virtual sensor to mimic real-time decoding. To verify that decoders predicted position with far better than chance accuracy, error performance was compared for decoders trained on calcium traces that were aligned versus misaligned with position tracking data (panel C, Supplement to *Figure 3*). Position decoding performance was compared for different methods of extracting online fluorescence traces including *contour-based* (CB) versus *contour-free* (CF) methods as explained below.

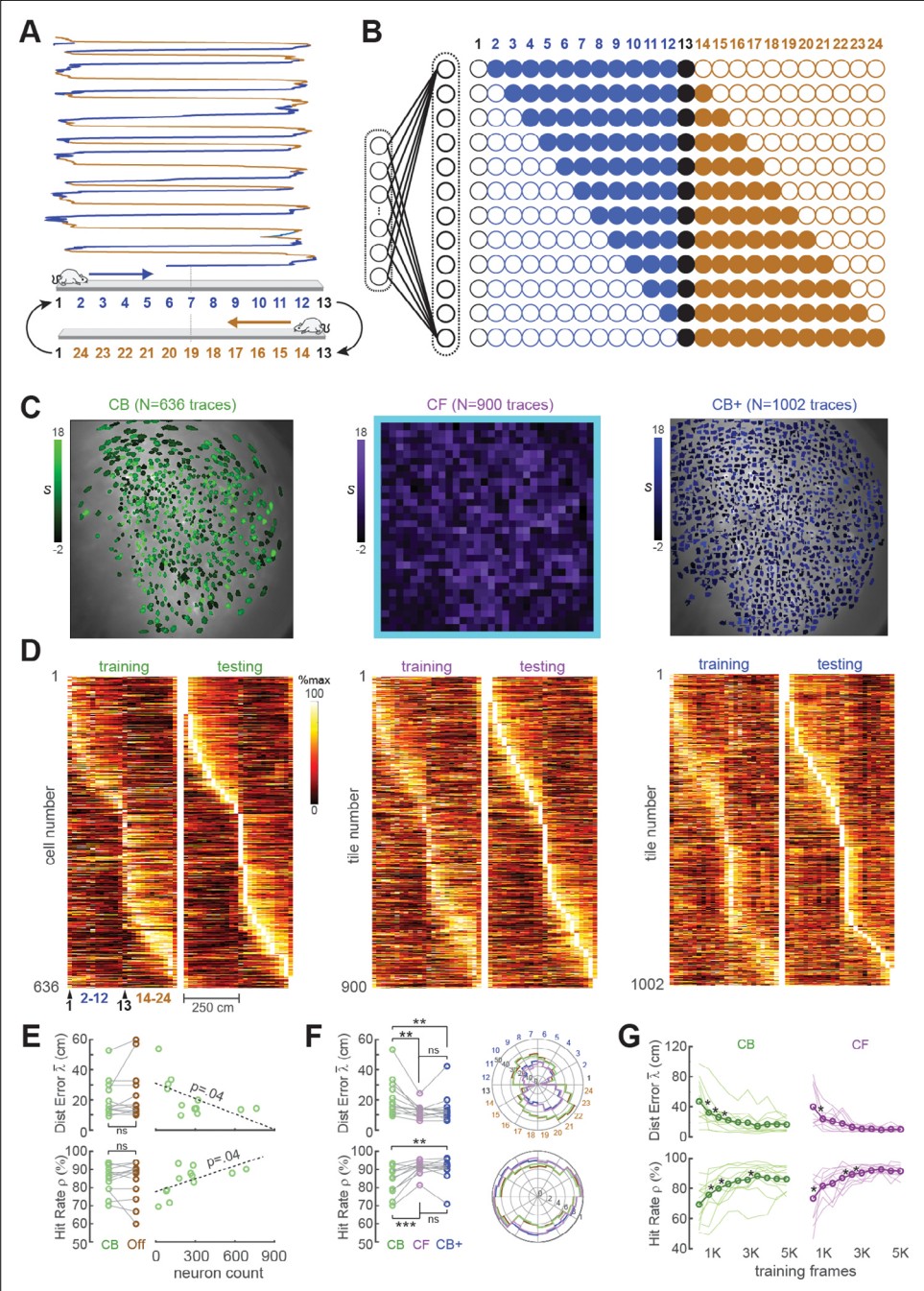

**Figure 3.** Position decoding from CA1 cells. (**A**) The CA1 layer of the hippocampus was imaged while rats ran laps on a 250-cm linear track; for position decoding, the rat's path was circularized and subdivided into 24 position bins, each ~20 cm wide. (**B**) $N$ linear classifer inputs (one per calcium trace) were mapped to 12 outputs using a Gray code representing the 24 track positions (open/filled circles show units with target outputs of $-1/+1$ at each position). (**C**) Regions of interest (ROIs) from which contour-based (CB, left), contour-free (CF, middle), and expanded contour-based (CB+) calcium traces were extracted in Rat #6; ROI shading intensity shows the similarity score, $S$, for traces extracted from the ROI. (**D**) Tuning curve heatmaps from training and testing epochs for CB (left), CF (middle), and CB+ traces extracted from Rat #6; rows are sorted by location of peak activity in the testing epoch. (**E**) Decoding performance did not differ for CB versus offline (Off) traces (left graphs), but was significantly better for sessions with larger numbers of calcium traces (right graphs). (**F**) Decoding performance averaged over track positions within each session (left graphs) and over sessions at each circularized track position (right graphs). (**G**) Decoding performance as a function of training set size; asterisks mark significant improvement with the addition of 500 more frames to the training set. For all panels: *p < 0.05, **p < 0.01, ***p < 0.001.

*Figure 3 continued on next page*

*Figure 3 continued*

The online version of this article includes the following figure supplement(s) for figure 3:

**Figure supplement 1.** Decoding position from calcium traces on the linear track.

## CB trace decoding

CB trace extraction derived calcium traces by summing fluorescence within pixel masks defining ROIs overlapping with identified neurons (*Figure 3C*; *Video 2*). Neurons were identified during the intermission period by using the CaImAn pipeline (*Giovannucci et al., 2019*) to perform constrained nonnegative matrix factorization (CNMF) (*Giovannucci et al., 2019*; *Pnevmatikakis et al., 2016*) on the training dataset and extract ROIs for demixed calcium traces. Running CaImAn on 4K–5K frames of training data required 30–60 min of computing time on the host PC, making this the slowest step performed during the intermission period (see *Figure 1B*). After contour ROIs had been identified, an additional ~5 min of computing time was required to feed all of the training data frames through a simulator to reconstruct online calcium traces that would have been extracted in real time during the first half of the session using the extracted contour masks. The simulated online traces were then aligned frame-by-frame with position data and used to train a linear classifier for decoding the rat's position on the linear track from calcium activity; training the decoder required a maximum ~60 s of computing time.

Accurate position decoding requires that calcium trace inputs to the decoder must retain stable spatial tuning across the training and testing epochs of the session. To verify that this was the case, spatial tuning curves were derived for CB calcium traces during the first (training) versus second (testing) half of each session (*Figure 3D*, left). Analyses of spatial selectivity and within-session spatial tuning stability (panels D, E in Supplement to *Figure 3*) indicated that large proportions of CB traces in CA1 exhibited stable spatial tuning between the training and testing epochs, as required for accurate decoding. The real-time decoder was trained and tested on CB traces from all ROIs identified by CaImAn in each session; we did not restrict the decoder's training set to a subset of traces that met a spatial selectivity criterion.

Two measures were used to quantify accuracy of position predictions decoded from calcium traces in each frame: *Distance error* $\lambda(t)$ was the absolute distance (in cm) between the true versus decoded position in frame $t$; the mean distance error over frames in the testing epoch shall be denoted $\bar{\lambda}$. *Hit rate* $\rho$ was the percentage of all frames in the test epoch for which the predicted position was within ±30 cm of the true position. *Figure 3E* shows that when the decoder was trained and tested with DeCalciOn's online (non-demixed) CB traces, there was no significant difference in mean decoding

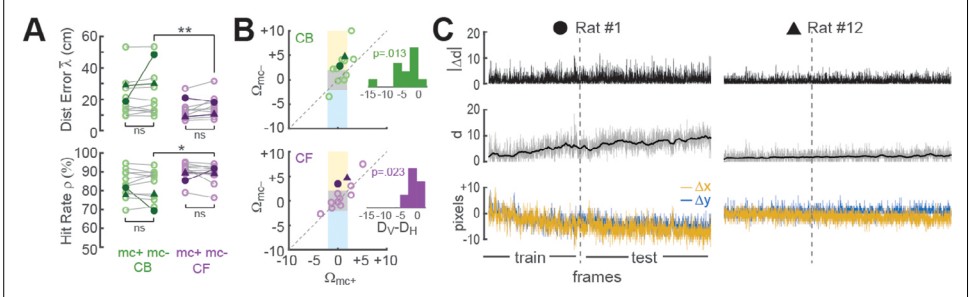

**Figure 4.** Improvement of place cell decoding by online motion correction. (**A**) Session-averaged performance of decoding from CB and CF traces is compared with (mc+) versus without (mc−) online motion correction; filled symbols '●' and '▲' mark data from example sessions shown in panel 'C'. (**B**) Scatter plots compare motion artifact scores with (Wmc+) versus without (Wmc−) online motion correction (see main text for explanation of shaded regions); insets show that points lie significantly nearer to the vertical ($D_V$) than horizontal ($D_H$) midline indicating a significant benefit of online motion correction. (**C**) Displacement of sensor image against reference alignment plotted over all frames in the session (*x*-axis) for Rat #1 ('●') which accumulated a large (~10 pixels) shift alignment error over the session, and Rat #12 ('▲') which exhibited transient jitter error but did not accumulate a large shift error; bottom graphs show signed horizontal ($\Delta x$) and vertical ($\Delta y$) displacement from the reference alignment, middle graphs show distance ($d$) from reference alignment. And top graphs show absolute value of time derivative of distance ($|\Delta d|$) from the reference alignment. *p<.05; **p<.01.

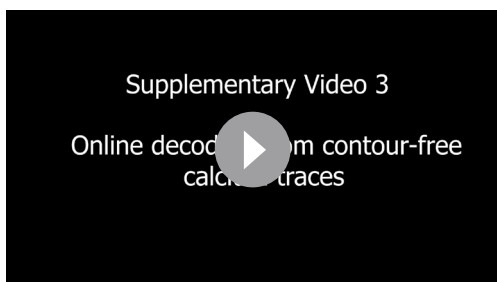

**Video 3.** Real-time decoding of contour-free calcium traces. The online image display (left window) shows the motion corrected and enhanced mosaic of contour-free pixel mask tiles. The right window shows a scrolling heatmap display of 103 (out of the total 900) calcium traces with the highest tuning curve similarity scores, *S* (see main text). The calcium trace rows in the heatmap are sorted by the peak activity location of each trace, so that calcium activity can be seen to propagate through the population as the rat runs laps on the linear track. The line graph at the top shows the rat's true position (blue line) together with its predicted position (orange line) decoded in real time by the linear classifier. For demonstration purposes, trace intensities shown in the scrolling heatmap are normalized within the ranges of their own individual minimum and maximum values.

https://elifesciences.org/articles/78344/figures#video3

accuracy ($\bar{\lambda}$ : $t_{11}$ = −1.32, p = 0.21; $\rho$: $t_{11}$ = 1.32, p = 0.21) from when the same decoder was trained on demixed offline ('Off') traces extracted by CalmAn from the same set of ROIs. Hence, decoding from online (real-time) CB traces versus demixed offline traces yielded similarly accurate position predictions. Unsurprisingly, the accuracy of decoding position from calcium traces was proportional to the population size of detected neurons ($\bar{\lambda}$ : R = −0.6, p = 0.0375; $\rho$: $R_1$ = 0.59, p = 0.0426; *Figure 4E*, right). Hence, the accuracy of position predictions improves with the number of calcium traces from which predictions are generated.

## CF decoding

In contrast with CB extraction, CF extraction derived calcium traces by summing fluorescence within pixel masks that did not overlap with identified neurons. Instead, ROIs were obtained by partitioning the 512 × 512 image into a 32 × 32 sheet of tiles, each tile measuring 16 × 16 pixels in area (*Video 3*); 124 tiles bordering the frame (shaded light blue in middle plot of *Figure 3C*) were excluded to eliminate edge contamination by motion artifacts. The CF extraction method always yielded a total of 900 calcium traces per rat (one for each square tile). By contrast, the CB method (see above) yielded a variable number of traces depending upon how many neuron ROIs were detected in the training dataset by CalmAn.

Unlike CB trace extraction, CF extraction does not require any computing time for contour identification or trace simulation during the intermission period, because the decoder is trained directly on 900 traces of CF data stored to memory during the training epoch; this training consumed <60 s of computing time on the host PC for 5K frames of training data. Hence, CF trace extraction required much less computing time during the intermission period than the CB trace extraction (which requires 30–60 min of computing time during intermission to identify trace ROIs, as explained above). Since CF calcium traces were derived from square tiles that were not aligned with locations of individual neurons in the sensor image, each CF trace could be modulated by fluorescence from multiple neurons within the ROI of its assigned tile. CF traces are thus analogous to multiunit spike recordings in neurophysiology that combine signals from multiple neurons. Similarly, CB traces can be regarded as analogous with single-unit recordings, since they are better at isolating signals from single neurons.

Like CB traces, CF traces in CA1 also exhibited stable spatially selective tuning (*Figure 3D*, middle). However, individual CF traces showed less within-session stability and fired over wider regions of the track (panels D, E in Supplement to *Figure 3*) than individual CB traces, as should be expected if CF traces contain signals from multiple place cells and CB traces isolate signals from individual place cells. But at the population level, decoding the rat's position from CF traces was as or more accurate than decoding from CB traces. As in the case of CB traces, the real-time decoder was trained and tested on all CF traces (*N* = 900 per rat in the tile mosaic) regardless of their spatial selectivity. When the accuracy of CF versus CB decoding was compared by performing paired *t*-tests on the $\bar{\lambda}$ and $\rho$ measures defined above (*Figure 3F*, left), CF decoding was more accurate than CB decoding ($\bar{\lambda}$ : $t_{11}$ = −3.43, p = 0.0056; $\rho$: $t_{11}$ = 4.6, p = 7.49e$^{-4}$). Analogous results have been reported in electrophysiology, where it has been found that decoding from multiunit spike trains can be more accurate than decoding from sorted spikes sourced to single units (*Deng et al., 2015*).

Since decoding accuracy for CB traces was proportional to population size (*Figure 3E*, right), it is possible that the reason CF traces provided better decoding accuracy was because they were

greater in number (in all rats, there were more CF than CB traces). To test this, CaImAn's sensitivity parameters were adjusted to detect more CB traces in each rat, yielding an expanded population (CB+) of CB traces that was larger than the original population of CB traces in each rat (*Figure 3C*, right). *Figure 4E* shows that CB+ decoding was more accurate than CB decoding ($\bar{\lambda}$ : $t_{11}$ = −3.09, p = 0.0013; $\rho$: $t_{11}$ = 3.7, p = 0.0035), but did not differ significantly in accuracy from CF decoding ($\bar{\lambda}$ : $t_{11}$ = −1.08, p = 0.3; $\rho$: $t_{11}$ = 0.83, p = 0.42). Hence, CF and CB+ traces yielded similar decoding accuracy, suggesting that the number of calcium trace predictors was a strong determinant of decoding accuracy. Deriving CB+ traces required about the same amount of computing time (30–60 min) during the intermission period as CB traces to identify contours and simulate online traces (see *Figure 1B*). Therefore, despite the similar accuracy of CF and CB+ traces, it was much more efficient to use CF traces since they did not require extra computing time to identify ROIs during the intermission period.

## Duration of the training epoch

An initial dataset must be collected at the beginning of each session to train the decoder before real-time imaging begins. The training epoch should be long enough to obtain sufficient data for training an accurate decoder, but short enough to leave sufficient remaining time in the session for real-time imaging. To analyze how position decoding accuracy on the linear track depended upon the duration of the training epoch, we varied the length of the training epoch in increments of 500 frames (see Methods). As expected, position decoding accuracy (quantified by $\bar{\lambda}$ and $\rho$) improved with the duration of the training epoch (*Figure 3G*). On average, asymptotic accuracy occurred when the training epoch reached ~3000 frames, beyond which increasing the duration of the training epoch yielded diminishing benefits for decoding accuracy.

## Online motion correction

Online motion stabilization (Stage 1) is the most computationally expensive stage in the real-time image processing pipeline, consuming more MPSoC hardware resources than any other stage (*Chen et al., 2022a*). To quantify the benefits of these high resource costs, we compared decoder performance with (mc+) and without (mc−) motion correction using CB or CF methods for CA1 calcium trace extraction (*Figure 4*). For these analyses, the decoder was trained on mc+ or mc− traces from the first 3000 frames in each session, and then tested on the corresponding mc+ or mc− traces from the remainder of the session.

Effects of motion correction upon session-averaged distance error and hit rate were computed using the formulas $\Delta\bar{\lambda} = \bar{\lambda}_{mc-}mc+$ and $\Delta\rho = \rho_{mc+}−\rho_{mc-}$, respectively (note that both formulas yield a positive result for cases where motion correction improves accuracy, and a negative result for cases where motion correction degrades accuracy). Paired *t*-tests found no difference between mc+ versus mc− conditions for $\Delta\bar{\lambda}$ or $\Delta\rho$ (*Figure 4A*), but session-averaged decoding accuracy may not be the most appropriate measure for motion correction's benefits because image stabilization is only beneficial when the image is translationally displaced from its reference alignment with the sensor, and this only occurs during a subset of frames within a session. To further assess the benefits of online motion correction, we analyzed how image stabilization reduced two distinct types of alignment error that can arise from translational displacement of the image on the sensor: *shift error* and *jitter error*.

### Shift error

Shift error occurs when the image permanently changes its steady-state alignment with the miniscope sensor (either suddenly or gradually) during the course of a session. For example, this can occur if the miniscope shifts its seating in the baseplate during a session. Significant shift error occurred in only 1 of the 12 linear track sessions analyzed for our performance tests (Rat #1 in *Figure 4C*); this session is marked by filled circles in panels of *Figure 4*. For this one session (but no others), benefits of motion correction were apparent even in the session-averaged data (*Figure 4A*). Hence, online motion stabilization was effective at correcting shift errors in the one session where they occurred; but given that shift errors occurred so rarely, it might be argued that correcting these errors does not adequately justify resource costs of online motion stabilization, especially if shift errors can be effectively prevented by other means such as rigidly attaching the miniscope to the baseplate.

### Jitter error

Jitter error occurs when the animal's inertial head acceleration causes transient motion of the brain inside the skull cavity, resulting in phasic translation of the brain image across the sensor. By definition, jitter error is temporary and lasts only until the brain image returns to its original alignment with the sensor after inertial motion stops. Jitter error can occur when the animal jerks or angles its head in such a way that the brain moves within the skull. Motion of the brain inside the skull is a natural phenomenon during free behavior, and for this reason, jitter error cannot easily be prevented by means other than online motion stabilization. To analyze whether online stabilization was effective at correcting for jitter error, we used the distance formula to compute displacement of each frame's non-motion-corrected image against the reference alignment: $d_t = \sqrt{\Delta x_t^2 + \Delta y_t^2}$ . Two Pearson correlation coefficients ($R_{mc+}$ and $R_{mc-}$) were derived for each session by correlating the distance error vectors ($\lambda_{mc+}$ and $\lambda_{mc-}$) with the displacement vector **d** over all frames in the testing epoch for the session. Positive $R$ values indicated that distance error was larger during frames that were misaligned on the sensor, as would be expected if decoding accuracy were impaired when jitter artifact corrupted the values of calcium traces. Conversely, negative $R$ values indicated that decoding error was *smaller* during frames that were misaligned on the sensor, which might occur if jitter error was correlated with the rat's position on the track in such a way that the decoder artifactually learned to predict the rat's location from image displacement rather than from neural calcium activity. $R$ values from each session were converted into motion artifact error scores using the formula $\Omega = -\log_{10}(P) \times \text{sign}(R)$, where $R$ is the correlation between distance error and image displacement and $P$ is the significance level for $R$; the p < 0.01 significance level for the similarity score is $S > 2$, since $\log_{10}(0.01) = 2$.

*Figure 4B* shows scatterplots of $\Omega_{mc+}$ versus $\Omega_{mc-}$ derived from CB (top) or CF (bottom) traces for all 12 linear track sessions in the performance testing analysis. Sessions falling within the central gray square region (3/12 for CB traces, 6/12 for CF traces) had non-significant motion artifact error scores before as well as after online motion correction, indicating that for these sessions, there was little benefit from online motion stabilization. Sessions falling within the upper yellow region (6/12 for CB traces, 2/12 for CF traces) had significant positive motion artifact error scores in the mc− condition that resolved in the mc+ condition, indicating that in these sessions, online motion correction successfully prevented degradation of decoding accuracy by motion artifact. Sessions falling within the lower blue region (1/12 for CB traces, 0/12 for CF traces) had significant 'negative' motion artifact error scores in the mc− but not mc+ condition, suggesting that without motion stabilization, the decoder artifactually learned to predict position from location-dependent motion artifacts rather than from neural calcium activity; this ersatz decoding was reduced by online motion correction. Further examples of decoders learning to predict behavior from image motion are shown below (see 'Decoding instrumental behavior from OFC neurons').

In *Figure 4B* and a small number of plotted points fell outside any of the shaded regions indicating that significant motion artifact scores occurred in both the mc− and mc+ conditions. However, the mean distance of points from the vertical midline ($D_V$) was significantly smaller than their mean distance ($D_H$) from the horizontal midline in scatterplots of both CB ($t_{11} = 2.63$, p = 0.0232) and CF ($t_{11} = 2.92$, p = 0.0139) traces. This indicates that correlations (positive and negative) between distance error and motion artifact were significantly reduced by online motion correction of both CB and CF traces. Hence, online motion stabilization benefitted decoding accuracy by improving the accuracy of online predictions for frames with motion artifact.

## Decoding instrumental behavior from OFC neurons

Another set of performance tests was carried out using the V4 UCLA Miniscope to image neurons in OFC while Long-Evans rats (*n* = 2) performed a 2-choice instrumental touchscreen task (*Figure 5A*). Each trial of the task began with the appearance of a central fixation stimulus (circle) on the screen, which the rat was required to touch. After touching the center stimulus, two visually distinct choice stimuli (A and B) appeared on the left and right sides of the screen. Assignment of stimuli A/B to the left/right side of the screen was randomized on each trial, but the correct response was always to touch the stimulus on one side (left for JL66, right for JL63) regardless of its identity. A correct response immediately triggered audible release of a food pellet into a magazine located behind the rat; an incorrect response triggered a 5-s timeout period. After each rewarded or non-rewarded outcome, a 10-s intertrial interval was enforced before initiating the next trial. *Figure 5B* (top) summarizes

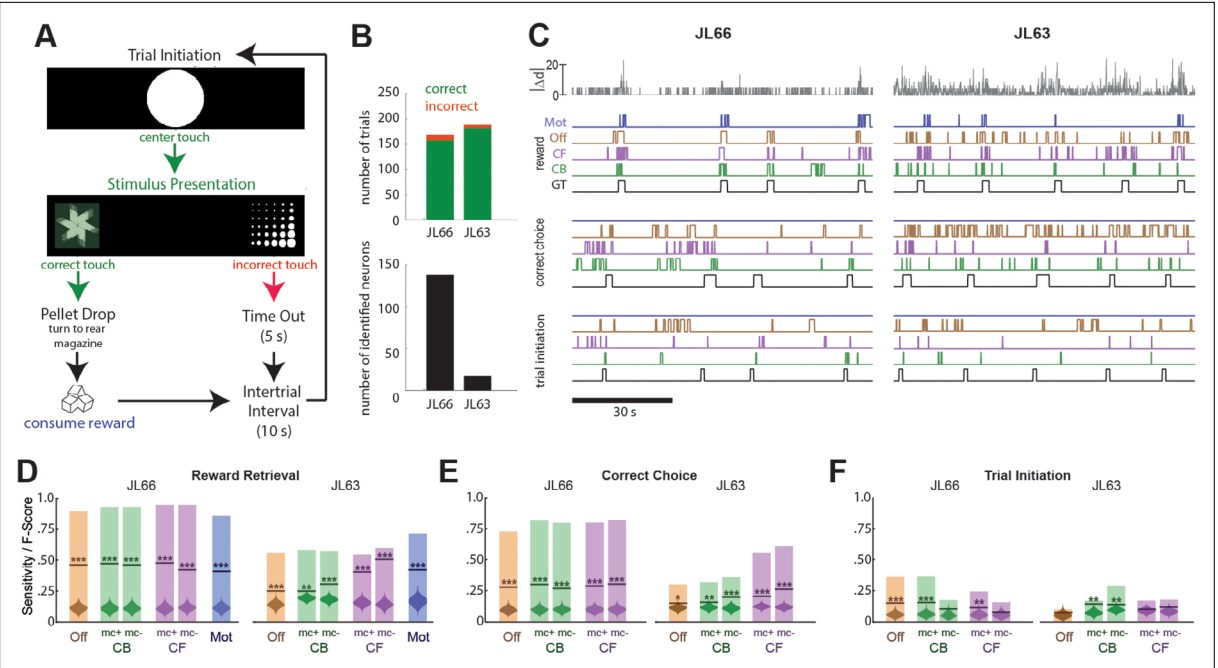

**Figure 5.** Decoding instrumental behavior from orbitofrontal cortex (OFC) calcium activity. (**A**) OFC neurons were imaged while rats performed a 2-choice touchscreen task (see main text). (**B**) Number of trials (top) and number of neurons identified by constrained non-negative matrix factorization (CNMF) (bottom) during the two analyzed sessions (JL66 and JL63). (**C**) Jitter error $|\Delta d|$ (gray traces, top) and real-time predictions of behavior labels (reward retrieval, correct choice, trial initiation) are shown for 90-s periods from the testing epoch of each rat (JL66 and JL63) after the binary tree decoder had been trained on 100 trials of offline (Off), contour-based (CB), or contour-free (CF) calcium traces; black traces show ground truth (GT) behavior labels, blue traces show predictions derived from motion vectors only (Mot). Sensitivity (shaded bars) and *F*-scores (horizontal lines) are shown for binary classification of reward retrieval (**D**), correct choice (**E**), and trial initiation (**F**) events using decoders trained on Off, CB, CF, or Mot predictors; for online CB and CF predictors, classifer performance is plotted separately for traces derived with versus without motion correction (mc+ vs. mc−), respectively. Violin plot inside each bar shows distribution of *F*-scores obtained from 1000 shuffles of event labels; asterisks over actual *F*-scores indicate significant difference from the shuffle distribution (***p < 0.001, **p < 0.01, *p < 0.05).

behavioral performance during 45-m imaging sessions (one for each rat) conducted after training on the discrimination task. In both sessions, rats performed >150 trials with >95% correct choices.

## Predicting behavior from calcium traces

Calcium traces were extracted from image data using each of the three methods described above: online contour-based (CB), online contour-free (CF), or offline contour-based (Off) extraction. *Figure 5B* (bottom left) shows that the number of orbitofrontal neurons (and thus the number of CB/Off traces) identified by CaImAn was much higher in one rat (JL66, *N* = 138 cells) than the other (JL63, *N* = 17 cells), whereas the number of CF traces was *N* = 900 in both rats (for reasons explained above).

To decode behavior from calcium traces, image frames were classified into five mutually exclusive categories: (1) *pre-trial* frames occurring within a 1-s window prior to each touch of the central fixation stimulus, (2) *correct choice* and (3) *incorrect choice* frames occurring between touches of the central fixation stimulus and touches of the correct or incorrect choice stimulus, respectively, (4) *reward retrieval* frames occurring within a 2-s window after reward magazine entry, and (5) *intertrial* frames not belonging to any of the previous four categories. A binary tree classifier was trained to predict behavior category labels from calcium trace vectors extracted using the CB, CF, or Off methods described above. As a control, the classifier was also trained to predict the behavior category from image motion displacement (Mot) rather than calcium traces (see Methods). For each session, the classifier was trained on data from the first 100 trials and then tested on the subsequent 50 trials. During testing, the predicted behavior label for each frame was taken to be the label that had been selected by the decoder most frequently within a five-frame window that included the current and previous four frames. Performance was assessed using two measures: *sensitivity*, the percentage of behavior events during the test epoch that were correctly predicted, and *F*-score, a standard measure

of binary classification accuracy that weighs sensitivity against precision of the classifier (see Methods). *Figure 5C* shows 90 s of example data from the testing epoch of each session (J66 and J63). In the figure, traces of ground truth behavior categories are aligned to real-time output from decoders trained on one of the four predictors (Off, CB, CF, and Mot).

Performance of the real-time decoder was assessed for predicting three of the five behavior categories: reward retrieval, correct choice, and trial initiation. Decoding performance was not analyzed for the 'incorrect choice' behavior category because very few incorrect choice trials occurred during the analyzed sessions, nor was it analyzed for the 'intertrial' category because rats exhibited variable behavior during intertrial intervals, reducing predictive validity.

### Reward retrieval

*Figure 5D* shows performance for real-time decoding of reward retrieval events from each of the four predictors (Off, CB, CF, and Mot). Sensitivity for detecting reward retrieval was higher for Rat JL66 than JL63, which is not surprising since predictions were derived from a larger number of detected neurons in Rat JL66. In both rats, sensitivity was similar when the decoder was trained on Off, CB, or CF traces. The *F*-score was higher for CF than Off or CB traces in Rat JL63 but not JL66, suggesting that in cases where a small number of neurons are detected (as in Rat JL63), decoder precision can be improved by using CF rather than CB traces. As discussed above for CA1 decoding, this benefit is likely to accrue from the larger number of CF than CB traces in cases where the detected neuron count is low.

### Correct choice

*Figure 5E* shows performance for real-time decoding of correct choice events from each of the four predictors (Off, CB, CF, and Mot). Sensitivity was higher for Rat JL66 than JL63, which again is not surprising since predictions were derived from a larger number of neurons in Rat JL66. In rat JL66, sensitivity for detecting reward retrieval was similar regardless of whether the decoder was trained on Off, CB, or CF traces. However, in Rat JL63, sensitivity was higher for CF traces, again suggesting that in cases where a small number of neurons are detected (as in Rat JL63), decoder performance is better for CF than CB traces.

### Trial initiation

*Figure 5F* shows that real-time decoding of trial initiation events was much less accurate than for reward retrieval (*Figure 5D*) or correct choice (*Figure 5E*) events, even in Rat JL66 where there were a large number of detected OFC neurons. This result suggests that trial initiation events were not robustly encoded by OFC neurons; hence, decoding of trial initiation was not considered further.

## Online motion correction

In both rats, real-time decoding of reward retrieval and correct choice events was similarly accurate (from both CB and CF traces) regardless of whether online motion correction was used (mc+ vs. mc−; *Figure 5D–F*). This suggests that online motion correction did not benefit decoding accuracy. However, in both rats, jitter error (quantified by |*d*|) was significantly greater during reward retrieval than during other behaviors (*Figure 6A*), indicating that retrieval of food from the magazine hopper caused the brain to move inside the skull. In rat JL63 (but not JL66), |*d*| was also greater during correct choice events than during other behaviors (*Figure 6B*), indicating that pressing the touch-screen may also have sometimes caused the brain to move inside the skull. Given that brain motion was significantly correlated with some behaviors, it is important to consider the possibility that in the absence of motion correction, the decoder may have learned to predict behavior from motion artifacts rather than from calcium activity. If so, then even though prediction accuracy remains high without motion correction, it would still be necessary to perform motion correction to assure that the real-time decoder derives its predictions from calcium activity rather than motion artifacts.

To investigate this, we isolated all frames that occurred during a given type of behavior event (reward retrieval or correct choice), and then performed rank sum tests to compare the median value of |*d*| from frames in which the behavior event was correctly classified ('predicted') versus frames in which it was not ('unpredicted'). Without motion correction, the median jitter error |*d*| in both rats was significantly higher for frames in which CB traces correctly predicted reward retrieval (*Figure 6C*, top graphs) or correct choice (*Figure 6D*, top graphs); this effect was abolished after motion correction (*Figure 6C, D*; bottom graphs). Similarly, without motion correction the median |*d*| in both rats was

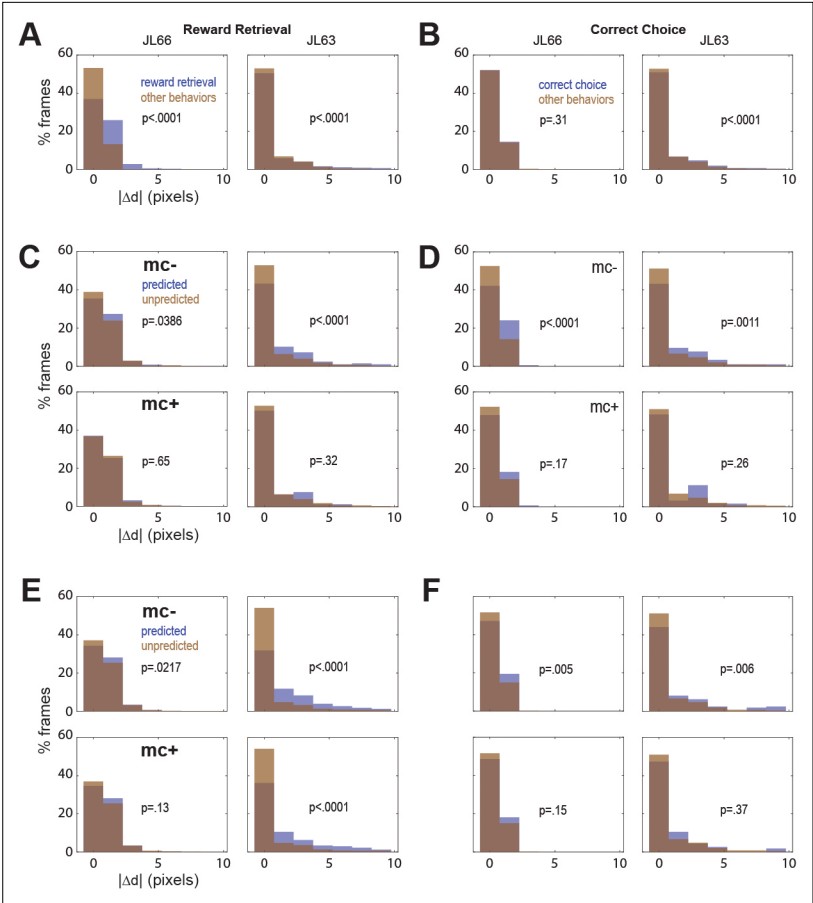

**Figure 6.** Correlation between motion artifact and decoding accuracy. (**A**) Jitter error $|\Delta d|$ is significantly greater during reward retrieval events than other behaviors in both rats. (**B**) $|\Delta d|$ is significantly greater during correct choice events than other behaviors in rat J63 but not JL66. (**C**) When CB traces are extracted without online motion correction (mc−; top), $|\Delta d|$ is significantly greater during accurately classified ('predicted') reward retrieval event frames than inaccurately rejected ('unpredicted') reward event frames; this difference is abolished when online motion correction is applied before trace extraction (mc+; bottom). (**D**) Same as 'C' but for predictions of correct choice events. (**E**) Same as 'C' but for CF rather than CB traces; note that for rat JL63 online motion correction does not abolish the correlation between decoding accuracy and motion artifact. (**F**) Same as 'D' but for CF traces extracted without (top) versus with (bottom) online motion correction.

significantly higher for frames in which CF traces correctly predicted reward retrieval (*Figure 6E*, top graphs) or correct choice (*Figure 6F*, top graphs); this effect was also abolished after motion correction (*Figure 6E, F*; bottom graphs) in all but one case (reward retrieval predictions in JL63).

## Real-time decoding latency

DeCalciOn's real-time feedback latency, $\tau_F$, is equal to the time that elapses between photons hitting the miniscope's image sensor and the rising edge of the TTL output signal that is triggered when the decoder detects a target pattern of neural activity. The total feedback latency is the sum of three delays that accrue sequentially: $\tau_F = \tau_L + \tau_T + \tau_I$, where $\tau_L$ is the light-gathering delay, $\tau_T$ is the frame transmission delay, and $\tau_I$ is the image processing delay.

## Light-gathering delay ($\tau_L$)

The first delay in the real-time decoding sequence arises from the time it takes for the sensor to gather light. Each image frame is formed on the sensor by summing light from photons that arrive during an exposure time window that is ~1 ms shorter than the frame interval, $I$. For example, at a frame rate of $F \approx 20$ Hz (as in experiments presented here) the frame interval is $I \approx 50$ ms, so the exposure time would be $(50-1) \approx 49$ ms, which is slightly shorter than $I$ because the sensor's circuits require ~1 ms

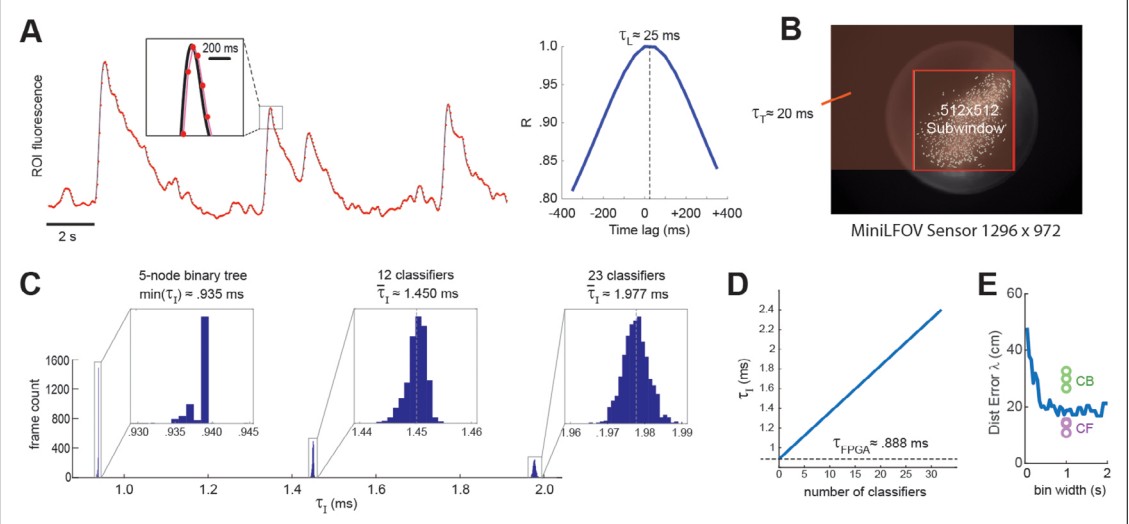

**Figure 7.** Real-time decoding latency. (**A**) Left graph illustrates how the light-gathering delay ($\tau_L$) is incurred as the sensor time integrates the true fluorescence signal (black line) by summing light gathered during each frame exposure, yielding a time series of binned fluorescence values (red dots) that are delayed from the original fluorescence signal by 1/2 the frame interval (50-ms frame interval was used for this example); right graph shows temporal cross correlogram between the source versus binned fluorescence signals, peaking at 1/2 the frame interval ($\tau_L$ = 25 ms). (**B**) Transmission delay ($\tau_T$) is proportional to the number of pixels, $P$, in the shaded red rectangular area between the upper left corner of the sensor image and the lower right corner of the 512 × 512 imaging subwindow. (**C**) Image processing delay ($\tau_I$) for 1204 calcium traces (maximum supported) is proportional to the number of units in the linear classifier's output layer; empirical distributions of $\tau_I$ are shown for linear classifiers with a single output unit (here $\tau_I$ is approximated as the minimum latency for a 5-node binary tree decoder), 12 output units, and 23 output units. (**D**) Linear fit to mean $\tau_I$ (y-axis) as function of the number of linear classifiers (x-axis); the y-intercept corresponds to the mean field-programmable gate array (FPGA) processing delay. (**E**) Blue line shows mean distance error (y-axis) for maximum likelihood decoding of a rat's position on a linear track from spike counts of 53 place cells (recorded by Hector Penagos in Matt Wilson's lab at MIT30) using time bins of differing size (x-axis) for spike counts; circles plot distance error for real-time decoding of CB and CF calcium traces from 3 rats in our dataset for which the number of CB traces was very close to 50 (thus matching the number of place cells in the single-unit vs. calcium trace decoding data). It can be seen that decoding from single-unit spikes was slightly more accurate than decoding from CB calcium traces and slightly less accurate than decoding from CF calcium traces in these three rats.

to perform analog-to-digital (A/D) conversion of pixel values at the end of each exposure. Hence, at each end-of-frame, the captured image integrates the intensity of light that has hit the sensor during the preceding frame interval. This binned averaging of light intensity is tantamount to convolving the 'true' brain fluorescence signal with a boxcar kernel of width $I$, effectively shifting the sampled fluorescence signal rightward in time from the true fluorescence signal by a delay of $\tau_L = I/2$ (*Figure 7A*). Thus, at the frame rate of 20 Hz used here, $\tau_L \approx 50/2 = 25$ ms.

## Frame transmission delay ($\tau_T$)

The second delay in the real-time decoding sequence arises from the time required to transmit frame data from the miniscope sensor to the DAQ, and then from the DAQ to the Ultra96 frame buffer. The miniscope sensor does not begin transmitting pixel data for a given frame until after A/D conversion is completed for all pixels in that frame, so there is no temporal overlap between $\tau_L$ and $\tau_T$. Each data pixel is serially transmitted from the sensor to the DAQ via the coax cable, and then immediately (with a delay of only one pixel clock cycle) deserialized and retransmitted over the flywire bus from the DAQ to the Ultra96 (see *Figure 1A*). Hence, the time required for a frame of data to travel from sensor to DAQ and then from DAQ to Ultra96 can be measured as a single transmission delay, $\tau_T$. The transmission time for an image frame can be computed as $\tau_T = P \times C$, where $C$ is the pixel clock period (60.24 ns for the V4 Miniscope, 16.6 ns for the LFOV Miniscope) and $P$ is the number of pixels per frame. Only pixels within the 512 × 512 imaging subwindow are used for real-time decoding, the effective number of pixels per frame is $P = W(B − 1) + R$, where $B$ and $R$ are the bottom-most pixel row and right-most pixel column of the selected 512 × 512 imaging subwindow, and $W$ is the width (in pixels) of the sensor image (red shaded region in *Figure 7B*). The size of the V4 UCLA Miniscope sensor is 608 × 608 pixels, so if the 512 × 512 subwindow is positioned in the center of the V4 sensor, the effective number of pixels per frame is p = 608 (608/2 + 512/2 − 1) + 608/2 + 512/2 = 340,432

and the transmission delay is $\tau_T$ = 340,432 × 60.24 = 20.5 ms. The size of the LFOV sensor is 1296 × 972 pixels, so if the 512 × 512 subwindow is positioned in the center of the LFOV sensor, then the effective number of pixels per frame is p = 1296 (1296/2 + 512/2 − 1) + 972/2 + 512/2 = 1,171,030 and the transmission delay is $\tau_T$ = 1,171,030 × 60.24 = 19.5 ms. Thus, with a frame rate of 20 Hz and an imaging subwindow located near the center of the sensor, the V4 and LFOV Miniscopes both incur transmission delays of ~20 ms.

## Image processing delay ($\tau_I$)

The final delay in the real-time decoding sequence arises from the time it takes to execute the real-time image processing pipeline on the Ultra96 (see *Figure 2*). As explained above, Stages 1–3 of image processing (online motion correction, background subtraction, and trace extraction) are performed in the fabric of the FPGA; after all three of these stages are completed, Stage 4 (linear classifier decoding) is performed by the MPSoC's ARM core. Hence, the image processing delay can be decomposed into two sequential and non-overlapping latencies, $\tau_I = \tau_{FPGA} + \tau_{ARM}$, where $\tau_{FPGA}$ is the FPGA latency incurred during Steps 1–3 and $\tau_{ARM}$ is the ARM core latency incurred during Stage 4. Here, we estimated these latencies empirically by measuring the time elapsed between arrival of the last pixel of a frame in the Ultra96 image buffer (start of the $\tau_{FPGA}$ interval) and the rising edge of the TTL pulse triggered by ARM core decoder output (end of the $\tau_{ARM}$ interval). It should be noted that the FPGA begins performing Step 1 of image processing (online motion correction) immediately after the lower right pixel of the 128 × 128 image processing subwindow (see yellow box in *Figure 2*) is received in the buffer, which is prior to arrival of the last pixel of the 512 × 512 subwindow. Consequently, Stage 1 is often finished by the time the last pixel of the 512 × 512 subwindow arrives (it may finish slightly later if the 128 × 128 window is located near the lower right corner of the 512 × 512 window). For this reason, our empirical estimate of $\tau_{FPGA}$ primarily measures the time required for the FPGA to perform Stages 2 and 3 after Stage 1 has been completed. *Figure 7C* plots empirically measured distributions of $\tau_I$ using 1204 calcium traces (the maximum number of predictors supported by DeCalciOn). The TTL latency for a linear classifier with a single output unit is approximated as the minimum observed latency for a 5-node binary tree decoder (corresponding to the case where the shallowest tree node produces a positive output); distributions are also shown for classifiers with 12 and 23 output units. We computed a linear fit to the empirical function $\tau_I = f(M)$, where M is the number of output units. The y-intercept of this function provides an estimate for the FPGA processing delay of $\tau_{FPGA} \approx 0.888$ ms; the ARM core processing delay then adds an additional ~47.3 µs per output unit to the FPGA delay.

Summarizing, DeCalciOn's real-time feedback latency is $\tau_F < \tau_L + \tau_T + max(\tau_I)$. For the UCLA Miniscope sampling at a frame rate of 20 Hz, this equates to $\tau_F$ < 25 ms + 20 ms + 2.5 ms = 47.5 ms. Since the frame interval is 50 ms at the 20 Hz frame rate, the TTL output signal generated for each frame will occur at least 2.5 ms before the last pixel of the subsequent frame arrives in the Ultra96 frame buffer. To put this feedback latency into the context of the rat's behavior, a freely behaving rat can travel ~7.5 cm in 47.5 ms if it is running at the maximum observed speed on the linear track of ~150 cm/s (see panel A in Supplement to *Figure 3*). Hence, if we write $x(t)$ to denote the position a rat occupies at the moment when a calcium trace vector arrives at the sensor, and $x(t + \tau_F)$ to denote the position the rat later occupies at the moment when TTL feedback is generated from the decoded calcium trace vector, the maximum distance error that can accrue between detection and decoding of the trace vector is $\Delta x = |x(t) - x(t + \tau_F)|$ = 7.5 cm.

*Figure 7E* shows that when a maximum likelihood decoder is used to predict a rat's position on a linear track from single-unit spike trains of 53 place cells (*Center for Brains Minds Machines, 2017*) at a 47.5-ms latency from the rat's true position, the distance error asymptotes at a minimum of ~20 cm for spike count time windows longer than ~0.5 s. It is difficult to directly equate the exponential decay time of a genetically encoded calcium indicator with the width of a square time window used for counting single-unit spikes, but if we assume that the GCaMP7s indicator used here integrates spikes with a time constant that is comparable to counting spikes in 1-s time window, then the accuracy of real-time position decoding from calcium traces appears to be slightly worse than decoding from single-unit spikes when CB traces are used, and slightly better than decoding from single-unit spikes when CF traces are used (dots in *Figure 7E*). This comparison was made for a subset of calcium imaging rats (n = 3) in which the number of detected CB traces was very similar to the number of place cells (n = 53) in the single-unit dataset (*Center for Brains Minds Machines, 2017*). Hence, real-time

decoding of calcium traces with DeCalciOn achieves distance errors that are within a similar range as those seen in offline analysis of place cell spike trains.

## Discussion

Here, we have demonstrated the DeCalciOn system's capabilities for online image processing, calcium trace extraction and decoding, and short latency triggering of TTL output pulses for closed-loop feedback experiments.

### Source identification

Most in vivo calcium imaging experiments utilize offline algorithms to analyze calcium trace data that has previously been collected and stored during a behavioral task. Popular offline analysis packages such as CaImAn (*Giovannucci et al., 2019*) and MIN1PIPE (*Deng et al., 2015*) employ sophisticated algorithms like CNMF (*Pnevmatikakis et al., 2016*) to demix crossover fluorescence and source calcium traces to single neurons. Accurate source identification is important for studies where investigating the tuning properties of individual neurons is a major aim of data analysis, but DeCalciOn is designed for real-time imaging experiments in which fast and accurate decoding of behavior from neural population activity takes precedence over characterizing single-neuron firing properties. For real-time decoding, the high computational costs of source identification may not yield proportional benefits to decoding accuracy. Supporting this, we found that linear classifiers trained on calcium traces extracted online (without demixing) predicted behavior just as accurately as those trained on traces extracted offline (with demixing) using the same set of ROIs (*Figure 3E*; *Figure 5D–F*). We also found that decoders trained on traces extracted from contour-free ROIs (which do not overlap with individual neurons) predicted behavior as or more accurately than traces extracted from contour-based ROIs that overlapped with identified neurons in the image frame (*Figure 3F and G*; *Figure 4A and B*; *Figure 5D–F*). Analogous results have been reported in electrophysiology, where 'spike sorting' algorithms for sourcing action potentials to single neurons incur high computational overhead costs that are comparable to the costs of sourcing calcium traces to single neurons with CNMF; decoding behavior from electrophysiological recordings of multiunit (unsorted) spikes is more computationally efficient than decoding from sorted spikes and often incurs no cost (or even a benefit) to decoding accuracy (*Deng et al., 2015*). Results presented here indicate that as in electrophysiology studies, the high computing cost of source identification in calcium imaging studies must be weighed against the benefits. For real-time imaging studies, contour-free decoding approaches can offer a strategy for optimizing computing efficiency without compromising decoding accuracy.

### Online motion correction

After source identification, the most computationally expensive stage in the image processing pipeline is motion stabilization. DeCalciOn reduces motion stabilization costs by using a motion correction subwindow (128 × 128 pixels, see *Figure 2*) that is considerably smaller than the 512 × 512 pixel imaging window (yet still large enough to achieve reliable motion correction). Even so, deriving each frame's motion displacement vector at millisecond latencies consumes more MPSoC hardware resources than any other stage within the image processing pipeline (*Chen et al., 2022a*). To quantify the benefits of these high resource costs, we compared decoder performance with and without motion correction. We found that online motion stabilization was effective at preventing two problems that can arise from instability of the brain image on the sensor. First, when decoding a rat's position on the linear track, online motion correction prevented decoding accuracy from becoming degraded during frames in which brain motion shifted the sensor image, as evidenced by the fact the positive correlations between decoding error and image displacement were reduced by online motion correction (*Figure 4*). Second, in the instrumental touch screen task where brain motion was significantly correlated with specific behaviors, the decoder learned to predict behavior from image motion (rather than calcium fluorescence) in the absence of motion correction; the decoder was prevented from learning these ersatz predictions when the online motion correction algorithm was applied to the decoder's training and testing datasets (*Figure 5D–F*; *Figure 6*).

Motion correction is a transient phenomenon and is therefore not needed during every image frame. Illustrating this, session-averaged accuracy for decoding position on the linear track was

not significantly improved by online motion correction (*Figure 4A*). However, for real-time feedback experiments, the session-averaged prediction accuracy is not the most important measure of decoder performance. What matters most is the accuracy of decoding during specific frames when it is important for feedback stimulation to be delivered or withheld. If such frames happen to overlap with motion artifact that interferes with decoding accuracy, then failure to perform motion correction can have disastrous consequences for feedback experiments, regardless of whether or not motion correction improves the 'average' decoding accuracy. Based on these considerations and the results of our performance tests, we conclude that the high resource costs of online motion correction are well worth the benefits they accrue by increasing the reliability of online decoding of neural population activity.

## Comparison with other real-time imaging platforms

Several previous online calcium fluorescence trace extraction algorithms have been proposed; some have been validated only by offline experiments showing that they can generate online traces that rival the quality of offline traces, and have not yet been performance tested on hardware platforms to assess their performance at real-time imaging or decoding of calcium traces (*Friedrich et al., 2017*; *Friedrich et al., 2021*; *Taniguchi et al., 2021*). Other systems have been performance tested in real time, as we have done here for DeCalciOn. For example, *Liu et al., 2021* introduced a system for real-time decoding with the UCLA miniscope that, unlike DeCalciOn, requires no additional hardware components because it is implemented entirely by software running on the miniscope's host PC. The system does not perform online motion stabilization, which is a significant drawback for reasons discussed above (see 'Online motion correction'). *Liu et al., 2021* showed that their system can implement a single binary classifier to decode calcium traces from 10 ROIs in mouse cortex at latencies <3 ms; this latency measurement appears to only include the decoder's execution time for generating predictions from calcium vectors (equivalent to Step 4 of our image processing pipeline) but not the light-gathering delay time (*Figure 7A*) or data buffering time. *Liu et al., 2021* did not report how their system's latency scales with the number of classifiers or the size/number of calcium ROIs, but under the assumption that serial processing time on the host PC scales linearly with these variables, decoding latencies of several seconds or more would be incurred for the decoding stage alone if 12–24 classifiers were used to predict behavior from 1204 calcium traces, which is much longer then the millisecond latencies achieved by DeCalciOn in performance tests presented here (see *Figure 7C, D*).

*Zhang et al., 2018* performed 2-photon calcium imaging experiments that incorporated real-time image processing on a GPU to detect neural activity and trigger closed-loop optical feedback stimulation in mouse cortex. The time required for calcium trace extraction increased linearly with a slope of ~0.05 ms per ROI (from a $y$-intercept of 0.5 ms; see panel C of Supplementary Fig. 6 in *Zhang et al., 2018*,), implying a total trace extraction time of ~52 ms for 1204 ROIs. Summing this latency with the light-gathering delay (~15 ms at the system's default 30-Hz frame rate), data buffering delay (reported to be ~2.5 ms), online motion correction delay (reported to be ~3.5 ms), and decoder execution time (~2 ms), it can be estimated that when decoding 1024 traces, *Zhang et al., 2018* system would incur a delay of ~75 ms between light arriving at the sensor and triggering of closed-loop feedback. This latency is only 25 ms (~50%) longer than DeCalciOn's 47.5-ms decoding latency under similar conditions (*Figure 7*), and could possibly be improved using parallelization or other efficiency strategies. Although latency performance is similar for DeCalciOn's FPGA-based pipeline and *Zhang et al., 2018* GPU-based pipeline, the two systems have different hardware compatibilities: DeCalciOn is designed for compatibility with head-free miniscopes that use the UCLA DAQ interface, whereas *Zhang et al., 2018* system is designed for compatibility with hardware and software interface standards used in benchtop multiphoton imaging systems.

## Summary and conclusions

DeCalciOn's low-cost, ease of use, and latency performance compare favorably against other real-time imaging systems proposed in the literature. We hope that by making DeCalciOn widely available to the research community, we can provide a platform for real-time decoding of neural population activity that will facilitate novel closed-loop experiments and accelerate discovery in neuroscience and

neuroengineering. All of DeCalciOn's hardware, software, and firmware are openly available through miniscope.org.

## Materials and methods

### Subjects

A total of 14 Long-Evans rats (6F, 6M for CA1 experiments, 2M for OFC experiments) were acquired from Charles River at 3 months of age. Subjects were singly housed within a temperature and humidity controlled vivarium on a 12-hr reverse light cycle. Surgical procedures began after a 1-week acclimation period in the vivarium, and recordings and behavioral experiments began around 5 months of age. All experimental protocols were approved by the Chancellor's Animal Research Committee of the University of California, Los Angeles, in accordance with the US National Institutes of Health (NIH) guidelines.

### Surgical procedures

#### Hippocampal surgeries

Subjects were given two survival surgeries prior to behavior training in order to record fluorescent calcium activity from hippocampal CA1 cells. During the first surgery, rats were anesthetized with 5% isoflurane at 2.5 l/min of oxygen, then maintained at 2–2.5% isoflurane while a craniotomy was made above the dorsal hippocampus. Next, 1.2 µl of AAV9-Syn-GCamp7s (Addgene) was injected at 0.12 µl/min just below the dorsal CA1 pyramidal layer (−3.6 AP, 2.5 ML, 2.6 DV) via a 10-µl Nanofil syringe (World Precision Instruments) mounted in a Quintessential Stereotaxic Injector (Stoelting) controlled by a Motorized Lab Standard Stereotax (Harvard Apparatus). Left or right hemisphere was balanced across all animals. One week later, the rat was again induced under anesthesia and four skull screws were implanted to provide stable hold for the subsequent implant. The viral craniotomy was reopened to a diameter of 1.8 mm, and cortical tissue and corpus callosal fibers above the hippocampus were aspirated away using a 27- and 30-gauge blunt needle. Following this aspiration, and assuring no bleeding persisted in the craniotomy, a stack of two 1.8-mm diameter Gradient Refractive INdex (GRIN) lenses (Edmund Optics) was implanted over the hippocampus and cemented in place with methacrylate bone cement (Simplex-P, Stryker Orthopaedics). The dorsal surface of the skull and bone screws were cemented with the GRIN lens to ensure stability of the implant, while the dorsal surface of the implanted lens was left exposed. Two to three weeks later, rats were again placed under anesthesia in order to cement a 3D printed baseplate above the lens. First a second GRIN lens was optically glued (Norland Optical Adhesive 68, Edmund Optics) to the surface of the implanted lens and cured with UV light. The pitch of each GRIN lens was ≤0.25, so implanting two stacked lenses yielded a total pitch of ≤0.5. This half pitch provides translation of the image at the bottom surface of the lenses to the top while maintaining the focal point below the lens. This relay implant enables access to tissue deep below the skull surface. The miniscope was placed securely in the baseplate and then mounted to the stereotax to visualize the calcium fluorescence and tissue. The baseplate was then cemented in place above the relay lenses at the proper focal plane and allowed to cure.

#### OFC surgeries

Infusion of AAV9-CaMKIIα-GCaMP6f (Addgene, #100834-AAV9) in OFC was performed using aseptic stereotaxic techniques under isoflurane gas (1–5% in O₂) anesthesia prior to any behavioral testing. Before surgery animals were administered 5 mg/kg s.c. carprofen (NADA #141-199, Pfizer, Inc, Drug Labeler Code: 000069) and 1 cc saline. After being placed in the stereotaxic apparatus (David Kopf; model 306041), the scalp was removed. The skull was leveled +2 and −2 mm A-P and M-L to ensure that bregma and lambda were in the same horizontal plane. Small burr holes were drilled in the skull above the target. Rats were infused with GCaMP6f virus in OFC (AP = +3.7; ML = ±2.5; DV = −4.6) and afterwards, implanted with a GRIN lens and affixed with a custom 3D printed lens cover during the same surgery. A total of 0.05 µl of GCaMP6f virus was infused at a rate of 0.01 µl per minute in the target region. After each infusion, 5 min elapsed before exiting the brain. Two to three weeks after virus infusion and lens implantation, rats were again placed under anesthesia to remove the lens cover and check for signal. After confirming signal, a miniscope baseplate was affixed with dental cement which enabled quick placement and removal of the miniscope in a freely moving rat.

## Linear track task

After rats had been baseplated, they were placed on food restriction to reach a goal weight of 85% *ad lib* weight and then began behavioral training. Time between the beginning of the surgical procedures and the start of behavior training was typically 6–8 weeks. Rats earned 20 mg chocolate pellets by alternating between two rewarded ends of a linear track (250 cm) during 15-min recordings beginning 5 days after baseplating. After receiving a reward at one end, the next reward had to be earned at the other end by crossing the center of the track. A webcam mounted in the behavior room tracked a red LED located on the top of the Miniscope and this video was saved alongside the calcium imaging via the Miniscope DAQ software with synchronized frame timestamps. These behavior video files were initially processed by custom python code, where all the session videos were concatenated together into one tiff stack, downsampled to the video imaging frame rate, the median of the stack was subtracted from each image, and finally they were all rescaled to the original 8-bit range to yield the same maximum and minimum values before subtraction. Background subtracted behavior videos were then processed in MATLAB. The rat's position in each frame was determined using the location of the red LED on the camera. Extracted positions were then rescaled to remove the camera distortion and convert the pixel position to centimeters according to the maze size. Positional information was then interpolated to the timestamps of the calcium imaging video using a custom MATLAB script.

## Instrumental touchscreen task

Before pretraining, rats were acclimated to a gentle hold on the lap of the experimenter sitting adjacent to the test chamber. This acclimation took place over at least 1 week prior to an imaging session wherein the rat was acclimated to unscrewing the cover of the baseplate on the rat's head, then placing the miniscope on the baseplate while on and imaging. After checking placement, the miniscope was screwed onto the baseplate. The rat was then placed in the operant chamber and the miniscope was plugged into the commutator. Pretraining schedules were then initiated that culminated in independent nosepokes to the touchscreen (*Harris et al., 2021*). Following criterion of 60 committed trials or more during a 45-min session, rats were advanced to the main instrumental task (*Figure 5A*). Rats were required to initiate a trial by touching the white graphic stimulus in the center screen (displayed for 40 s), and after initiation rats would be presented with two identical stimuli (i.e., fan or marble) on the left and right side of the screen (displayed for 60 s), that they were required to nosepoke as either the correct spatial side ($p_R(B) = 1$; rewarded with one sucrose pellet) or incorrect spatial side ($p_R(W) = 0$). Thus, rats were required to ignore the properties of the stimuli and determine the better-rewarded side.

## Histology

At the end of the experiment, rats were anesthetized with isoflurane, intraperitoneally injected with 1 ml of pentobarbital, then transcardially perfused with 100 ml of 0.01 M phosphate-buffered saline (PBS) followed by 200 ml of 4% paraformaldehyde in 0.01 M PBS to fix the brain tissue. Brains were sectioned at 40-µm thickness on a cryostat (Leica), mounted on gelatin prepared slides, then imaged on a confocal microscope (Zeiss) to confirm GFP expression and GRIN lens placement.

## Hardware

The DeCalciOn system is designed for use with UCLA Miniscope devices (*Cai et al., 2016*) and other miniscopes (*Scott et al., 2018*; *de Groot et al., 2020*; *Blair et al., 2021*) that use the UCLA DAQ interface. The DAQ hardware requires modification by soldering flywires to the PCB (see *Figure 1*); instructions for doing this are available at https://github.com/zhe-ch/ACTEV (*Chen, 2023* copy archived at swh:1:rev:aa6393d3bd2dd490aa5369e1f2677e85e8a64a82) and pre-modified DAQ boards are available at miniscope.org. DeCalciOn is implemented on an Avnet Ultra96 development board (available from the manufacturer at Avnet.com), which must be mated to our custom interface board available at miniscope.org (the interface can also be ordered from a PCB manufacturer with design files available at https://github.com/zhe-ch/ACTEV) (*Chen, 2023* copy archived at swh:1:rev:aa6393d3bd2dd490aa5369e1f2677e85e8a64a82). ACTEV compiled hardware (for online image stabilization, image enhancement, calcium trace extraction) and embedded software (for real-time decoding and ethernet communication, which runs under the FreeRTOS operating system) can be programmed onto the Ultra96 board by copying a bootable bitstream file to a MicroSD card and

then powering up the board with the card inserted. The V4 and MiniLFOV versions of the bitstream used for experiments reported here are available at https://github.com/zhe-ch/ACTEV., (*Chen, 2023* copy archived at swh:1:rev:aa6393d3bd2dd490aa5369e1f2677e85e8a64a82) Bitstream files for other miniscope models and Vivado HLS C source code for generating new custom bitstream files are available on request.

## Software

Real-time imaging and decoding are controlled by custom RTI software (available at https://github.com/zhe-ch/ACTEV) (*Chen, 2023* copy archived at swh:1:rev:aa6393d3bd2dd490aa5369e-1f2677e85e8a64a82) which runs on the host PC alongside standard DAQ software (available at miniscope.org) for adjusting the gain and focus of the Miniscope device. The DAQ software's default operation mode is to receive and display raw Miniscope video data from the DAQ via a USB 3.0 port (and store these data if the storage option has been selected), and also to receive and display raw behavior tracking video from a webcam through a separate USB port. At the start of a real experimental session, data acquisition by both programs (DAQ and RTI software) is initiated simultaneously with a single button click in the RTI user interface, so that Miniscope video storage by the RTI software and behavior video storage by the Miniscope DAQ software are synchronized to begin at exactly the same time. This allows behavioral data stored by the DAQ software to be aligned with Miniscope video and calcium trace data stored by the RTI software. During the intermission period between initial data acquisition and real-time inference, data stored by DAQ and RTI software are used to train the linear classifier on the host PC. Trained classifier weights are then uploaded to the Ultra96 for real-time decoding.

## Contour-based pixel masks

Pixel masks for contour-based trace extraction were derived using the CaImAn (*Giovannucci et al., 2019*) pipeline (implemented in python) to analyze motion-corrected sensor images from the training dataset (as noted in the main text, this took 30–60 min of computing time on the host PC). CaImAn is an offline algorithm that uses CNMF to isolate single neurons by demixing their calcium traces. The CNMF method is acausal so it cannot be used to extract traces in real time. But during offline trace extraction, CaImAn generates a set of spatial contours identifying pixel regions from which each demixed trace was derived, which ACTEV then uses as pixel masks for online extraction of contour-based traces. Once pixel masks were identified from the training data, motion-correct miniscope video from the training period was passed through an offline simulator that used ACTEV's causal algorithm for extracting calcium traces from contour pixel masks. This yielded a set of calcium traces identical to those that would have been extracted from the training data in real time by ACTEV. These simulated traces were then used as input vectors to train the linear classifier. Since the online traces are not generated by CNMF (and are thus not demixed from one another), they are susceptible to contamination from fluorescence originating outside of contour boundaries. However, this crosstalk fluorescence did not impair decoding since similar accuracy results were obtained by training the linear classifier on online or offline contour-based traces (*Figure 4F, G*).

## Contour-free pixel masks

To implement contour-free trace extraction, we simply partitioned the 512 × 512 image frame into a 32 × 32 sheet of tiles, each measuring a square of 16 × 16 pixels (*Figure 4B*). No traces were extracted from 124 tiles bordering the edge of the frame, to avoid noise artifacts that might arise from edge effects in the motion stabilization algorithm. Hence, a total of 1024−124 = 900 pixel mask tiles were used for contour-free calcium trace extraction. These traces were derived in real time and stored to the host PC throughout the initial data acquisition period, so they were immediately available for training the linear classifier at the start of the intermission. Consequently, an advantage of contour-free trace extraction is that the intermission period between training and testing is shortened to just a few minutes, because the lengthy process of contour identification is no longer required. A disadvantage of contour-free trace extraction is that contour tiles do not align with individual neurons in the sensor image. As reported in the main text, this lack of alignment between neurons and pixel masks did not impair (and often enhanced) position decoding; however, contour-free decoding does place limits upon what can be inferred about how single neurons represent information in imaged brain regions.

## Fluorescence summation

After pixel masks were created using one of the two methods (contour-based or contour-free) and uploaded to the FPGA, calculations for extracting calcium traces from the masks were the same. Each mask specified a set of pixels over which grayscale intensities were summed to obtain the fluorescence value of a single calcium trace:

$$T\left(f\right) = \sum_{i=1}^{P} p_i\left(f\right)$$

where $T(f)$ is the summed trace intensity for frame $f$, and $p_i(f)$ is the intensity of the $i$th pixel in the mask for frame $f$, and $P$ is the number of pixels in the mask. Each contour-free mask was a square tile containing $16 \times 16 = 256$ pixels (*Figure 4B*), whereas the size of each contour-based mask depended upon where CalmAn identified neurons in the image (*Figure 4A*).

## Drop filtering

The mean size of contour-based masks was 20–100 pixels (depending on the rat), which was an order of magnitude smaller than contour-free tiles. In a few rats, a small amount of jitter sometimes penetrated the online motion correction filter, causing the stabilized sensor image to slip by 1–2 pixels against stationary contour masks during 1–2 frames (see Video S1, line graph at lower right). This slippage misaligned pixels at the edges of each contour mask, producing intermittent noise in the calcium trace that was proportional to the fraction of misaligned pixels in the contour, which in turn was inversely proportional to the size of the contour, so that motion jitter caused more noise in traces derived from small contour masks than large contour masks. To filter out this occasional motion jitter noise from calcium traces, a *drop filter* was applied to traces derived from contour masks that contained fewer than 50 pixels. The drop filter exploited the fact that genetic calcium indicators have slow decay times, and therefore, sudden drops in trace fluorescence can be reliably attributed to jitter noise. For example, the GCamp7s indicator used here has a half decay time *O'Keefe and Dostrovsky, 1971* of about 0.7 s, so at a frame rate of ~20 Hz, a fluorescence reduction of more than 5% between frames can only arise from jitter artifact. The drop filter defines a maximum permissible reduction in fluorescence between successive frames as follows:

$$\delta = GCq$$

where $G$ is the maximum possible fluorescence intensity for any single pixel (255 for 8-bit grayscale depth), $C$ is the number of pixels in the contour mask for the trace, and $q$ is a user-specified sensitivity threshold, which was set to 0.9 for all results presented here. The drop-filtered calcium trace value for frame $f$ was given by:

$$F\left(f\right) = max\left[T\left(f\right), F\left(f-1\right) - \delta\right]$$

It should be noted that while drop filtering protects against artifactual decreases in trace values, it offers no protection against artifactual increases that might masquerade as neural activity events. However, motion jitter almost never produced artifactual increases in fluorescence for the small contours to which drop filtering was applied, because small contours were 'difficult targets' for patches of stray fluorescence to wander into during jitter events that rarely exceeded 1–2 pixels of slippage. Larger contours were slightly more likely to experience artifactual fluorescence increases during jitter events, but in such cases, the artifact was diluted down to an inconsequential size because it only affected a tiny fraction of the large contour's pixels. In summary, jitter artifact was highly asymmetric, producing artifactual decreases but not increases for small contours, and producing negligible artifact of any kind for large contours.

## Spike inference

Once trace values have been extracted (and drop filtered if necessary), ACTEV can apply a real-time spike inference engine to convert raw calcium trace values into inferred spike counts for each frame. Briefly, the spike inference engine measures how much the trace value in the current frame has changed from the prior frame, $T(f) - T(f-1)$, and compares this against a threshold value, $\Phi$, which is equal to 2.5 times the standard deviation of difference values between successive frames of the same trace in the training dataset. For the CA1 data reported here, we found that the linear classifier

was more accurate at decoding raw (unconverted) calcium traces than inferred spike counts (panel A of Supplementary to *Figure 3*). Decoding from raw calcium traces is not only more accurate but also more computationally efficient; therefore, the main text reports results obtained by decoding raw calcium trace values.

## Training classifiers

To train the linear classifier, data stored during the initial acquisition period by the DAQ and RTI software were analyzed during the intermission period by custom MATLAB scripts, available at https://github.com/zhe-ch/ACTEV, (*Chen, 2023* copy archived at swh:1:rev:aa6393d3bd2dd490aa5369e-1f2677e85e8a64a82). Different linear classifier architectures were used for position versus behavioral decoding, as explained below. Training vectors and target outputs were passed to MATLAB's *fitceoc* function (from the machine learning toolbox) to compute the trained linear classifier weights, which were then uploaded from the host PC to the Ultra96 via ethernet, so that the real-time classifier running under FreeRTOS in the ARM core could decode calcium traces in real time.

### Linear classifier for position decoding

The linear classifier had $N$ inputs and $M$ outputs, where $N$ is the number of calcium traces. In performance tests reported here, the output layer utilized a Gray coding scheme to represent each of the 24 binned track locations (*Figure 3B*). Under this representation scheme, the total number of output units is equal to $M = K/2$, where $K$ is the number of categories (position bins) to be encoded. Since the linear track was subdivided into $K = 24$ spatial bins, there were $M = 24/2 = 12$ binary classifier units in the decoder's output layer. Training data consisted of calcium trace input vectors (derived by either contour-based or contour-free methods described above) and target position output vectors (encoding the rat's true position) for each Miniscope frame. As a control condition, calcium traces and tracking data were circularly shifted against one another by 500, 1000, 1500, 2000, or 2500 frames before training, and the trained classifier was then tested on correctly aligned trace inputs and target position outputs. Accuracy results from all five shift values were averaged together to obtain the 'shift' decoding accuracies plotted in panel C of the Supplement to *Figure 3*.

### Binary tree classifier for behavior decoding

The binary tree classifier had $N$ inputs and $M$ outputs, where $M$ is the number of behavior categories to be decoded. In performance tests reported here, there were $M = 5$ behavior categories (*pre-trial*, *correct choice*, *incorrect choice*, *reward retrieval*, and *intertrial*) as defined in the main text. Calcium traces were extracted from each image frame using the CB, CF, or Off methods described above. As a control measure, the classifier was also trained to predict the behavior category from image motion displacement (Mot) rather than calcium traces; the Mot predictor was a vector comprised from five elements per frame: $\Delta x$, $\Delta y$, $d$, $\Delta d$, and $|\Delta d|$ (see main text for definitions of symbols). For each session, the classifier was trained on data from the first 100 trials (40,998 frames for rat JL66, 44,877 frames for rat JL63) and then tested on the subsequent 50 trials (25,798 frames for rat JL66, 24,974 frames for rat JL63).

## Decoding calcium traces

Once classifier weights had been uploaded to the Ultra96 from the host PC, it was then possible for the embedded ARM core in the Ultra96 MPSoC to read in calcium trace values from the DRAM in real time as each frame of data arrived from the Miniscope, and present these values as inputs to the linear classifier. The ARM core was programmed in C++ to perform calculations equivalent to MATLAB's *predict* function (from the machine learning toolbox) and thereby generate the linear classifier output vector. For predicting the rat's position on the track from CA1 activity, the decoded position bin was taken to be that represented by the Gray code vector that most closely matched the classifier output vector. For categorizing the rat's behavior from OFC activity, the predicted behavior label was taken to be the label that had been most often selected by the decoder during a five-frame window that included the current and previous four frames. OFC decoding performance was assessed by computing *sensitivity* (the overall percentage of behavior events during the test epoch that were correctly predicted) and *F*-score (a standard measure of binary classification accuracy computed as $2PS/(P + S)$, where $S$ is the sensitivity and $P$ is the precision defined as the number of correct positive

results divided by the number of all positive results). Decoder predictions were sent back to the host PC via ethernet for storage, and were also sent to a programmable *logic mapper* running on the ARM core, which converted the linear classifer's output vector into a pattern of square pulses generated at on of 8 TTL pinouts from the Ultra96, capable of driving closed-loop neural or sensory feedback via external devices such as lasers, electrical stimulators, or audiovisual display equipment.

### Spatial tuning curves

To generate spatial tuning curves for fluorescence traces, each trace was first referenced to zero by subtracting its minimum observed fluorescence value (over all frames) from the value in every frame: $Z(f) = T(f) - min(T)$, where $Z(f)$ is the zero-referenced trace value for frame $f$. The 250-cm linear track was subdivided into 24 evenly spaced spatial bins (12 each in the left-to-right and right-to-left directions; see *Figure 2A*). Fluorescence activity, $A$, in each bin $b$ was averaged by

$$A(b) = \sum_{i=1}^{B} Z(f_i)/B$$

where $B$ is the number of visits to bin $b$ and $f_i$ is the $i$th frame during which the rat was visiting bin $b$. The tuning curve vector $[A_1, A_2, …, A_{24}]$ was then normalized by dividing each of its elements by the value of the maximum element. Normalized tuning curve vectors were plotted as heatmaps in *Figure 2C, D*. Spatial similarity scores ($S$) were computed as described in the main text; $S$ distributions for each rat are shown in Extended Data *Figure 3*.

### Correlation between decoding error and image displacement

Two Pearson correlation coefficients ($R_{mc+}$ and $R_{mc-}$) were derived for each session by correlating the distance error vectors ($\lambda_{mc+}$ and $\lambda_{mc-}$) in each frame with the displacement vector $\mathbf{d} = \{d_1, d_2, ..., d_F\}$, where $F$ is the total number of frames in the session. A necessary condition for computing reliable $R$ values was that $\lambda$ and $d$ had to be sufficiently dispersed along their respective axes to detect a significant linear relationship between the variables. However, within any given session, there were often some position bins in which motion artifact either never occurred or was so small in magnitude that it was impossible to assess the impact of motion artifact upon decoding accuracy at that position. For this reason, the motion-accuracy correlation analysis was performed only on frames from selected position bins in which the maximum value of $d$ exceeded a threshold of 4 pixels, which was a large enough displacement to alter the values of CB trace values extracted from images without motion correction. All frames from position bins meeting this criterion (and no frames from position bins that did not) were included in the correlation analysis. Hence, analysis was restricted to those locations on the track where trace values (and thus distance error of decoded positions) could potentially be affected by motion artifact and excluded locations where no influence was possible because of a lack of motion artifact.

## Acknowledgements

The authors thank Xilinx Corporation for donation of the Ultra96 board. We also thank Jeffrey Johnson for technical assistance in designing the custom interface board, and Shiyun Wang and Ryan Grgurich for helpful comments on the manuscript. This work was supported by NSF NeuroNex 1704708 (HTB, PG, DA, JC), 1UF1NS107668 (DA), and MH122800 (AI, HTB).

## Additional information

### Competing interests

Alicia Izquierdo: Reviewing editor, *eLife*. The other authors declare that no competing interests exist.

## Funding

| Funder | Grant reference number | Author |
| --- | --- | --- |
| NSF NeuroNex | 1707408 | Peyman Golshani<br>Jason Cong<br>Daniel Aharoni<br>Hugh T Blair |

The funders had no role in study design, data collection, and interpretation, or the decision to submit the work for publication.

## Author contributions

Zhe Chen, Conceptualization, Resources, Data curation, Software, Formal analysis, Validation, Investigation, Visualization, Methodology, Writing - original draft, Writing - review and editing; Garrett J Blair, Resources, Data curation, Software, Formal analysis, Validation, Investigation, Visualization, Methodology, Writing - review and editing; Changliang Guo, Resources, Software, Validation, Methodology; Jim Zhou, Data curation, Software, Formal analysis; Juan-Luis Romero-Sosa, Data curation, Investigation; Alicia Izquierdo, Conceptualization, Funding acquisition, Writing - review and editing; Peyman Golshani, Conceptualization, Resources, Supervision, Funding acquisition, Methodology, Project administration, Writing - review and editing; Jason Cong, Conceptualization, Resources, Software, Formal analysis, Supervision, Funding acquisition, Validation, Methodology, Project administration, Writing - review and editing; Daniel Aharoni, Conceptualization, Resources, Software, Supervision, Funding acquisition, Validation, Investigation, Methodology, Writing - review and editing; Hugh T Blair, Conceptualization, Resources, Data curation, Software, Formal analysis, Supervision, Funding acquisition, Validation, Investigation, Visualization, Methodology, Writing - original draft, Project administration, Writing - review and editing

## Author ORCIDs

Garrett J Blair (ID) http://orcid.org/0000-0003-2724-8914
Alicia Izquierdo (ID) http://orcid.org/0000-0001-9897-2091
Daniel Aharoni (ID) http://orcid.org/0000-0003-4931-8514
Hugh T Blair (ID) http://orcid.org/0000-0001-8256-5109

## Ethics

This study was performed in strict accordance with the recommendations in the Guide for the Care and Use of Laboratory Animals of the National Institutes of Health. All of the animals were handled according to approved Institutional Animal Care and Use Committee (IACUC) protocols (#2017-038) of the University of California Los Angeles. The protocol was approved by the Committee on the Ethics of Animal Experiments of UCLA. All surgery was performed under deep isoflurane anesthesia, and every effort was made to minimize suffering, including administration of pre- and post-surgical analgesia.

## Decision letter and Author response

Decision letter https://doi.org/10.7554/eLife.78344.sa1
Author response https://doi.org/10.7554/eLife.78344.sa2

# Additional files

## Supplementary files

• MDAR checklist

## Data availability

All hardware, software, and firmware are openly available through miniscope.org and at https://github.com/zhe-ch/ACTEV, (copy archived at swh:1:rev:aa6393d3bd2dd490aa5369e1f2677e85e8a64a82).

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
