## [Editor Report]

This article presents a novel realtime processing pipeline for miniscope imaging which enables accurate decoding of behavioral variables and generation of feedback commands within less than 50ms. The efficiency of this important tool for experiments requiring close-loop interaction with brain activity is demonstrated based on compelling measurements in two experimental contexts. This pipeline will be useful for a wide range of questions in system neuroscience.

---

## [Decision Letter]

**Decision letter after peer review:**

Thank you for submitting your article "A hardware system for real-time decoding of in-vivo calcium imaging data" for consideration by *eLife*. Your article has been reviewed by 3 peer reviewers, including Brice Bathellier as Reviewing Editor and Reviewer #1, and the evaluation has been overseen by Laura Colgin as the Senior Editor.

Essential revisions:

1) Provide a clear estimate of position accuracy that can be achieved with a space metric and not with a classifier accuracy with space bins whose size is unclear. Compare position estimates to what would be obtained from electrophysiology based on the literature.

2) Please provide a spatial distribution of speed, acceleration, arrests, and a description of the number of place cells and of the size of the place fields. Related to that, it is crucial to provide a plot of the accuracy of spatial decoding against spatial location to appreciate the performance of the system in real conditions, knowing that in linear tracks place fields are not homogeneously distributed.

3) Improve the description of the methodology and of imaging speed with respect to displacement speed according to referee 2's comments. It is particularly important to explain how much exploration time is needed to train the real-time classifier.

4) Provide an evaluation of the localisation information available in corrected motion artefacts in order to rule out that residual motion artefacts do not provide spatial information.

5) Repeat the experiment in a more complex environment, such as a 2D arena, in order to demonstrate the versatility of the method.

6) Provide a direct comparison of real performances based DeCalciOn to two previous online calcium imaging algorithms. The technical innovations of this work would be better highlighted by directly testing all three of these algorithms, ideally on similar datasets.

7) Provide a quantification of the spatial error (across different locations, and running speeds) made in a virtual experiment in which the goal is to emit a TTL signal at a specific location of the animal. The difference with point 7 is that this quantification includes not only classifier inaccuracies but also the time delay of the real time device in relation to the speed of the animal.

*Reviewer #1 (Recommendations for the authors):*

– Provide a clear estimate of position accuracy that can be achieved with a space metric and not with a classifier accuracy with space bins whose size is unclear.

– Provide measurements in 2D and estimates of both location and head direction. How does it compare with electrophysiology? Are the conclusions about the lack of importance of cell body identification still holding for these conditions?

*Reviewer #2 (Recommendations for the authors):*

– There is not enough information about the behavior. Please provide a spatial distribution of speed, acceleration, and arrests. Is there an influence on the decoding accuracy?

– There is not enough information about the place cells. How many place cells? What is the spatial information of the cells? What are the size of the place fields?

– Some details are lacking or are not explicitly described. For instance, I think the following information should be easier to obtain in the article. The 2.5m linear track is split into 13 regions, leading to regions of roughly 20cm. In linear tracks rats run with an average speed of 70cm.s^-1^, with maximum speed going up to 100cm.s^-1^. This means that running rats will stay between 280ms to 400ms in each zone. The video frames are down-sampled to 15 frames/s leading to 66ms between each frame. This implies that 4 to 6 frames will correspond to the same location.

– A clearer description of the methodology might help the reader:

Here is a short overview of the methodology:

Technological methodology:

Step 1: Correct for translational movement of brain tissue. Aa 128x128 pixel area with distinct anatomical features was selected within the 512x512 cropped subregion to serve as a motion stabilization window;

Step 2: ACTEV removes background fluorescence from the 512x512 image;

Step 3: ROI definition: The enhanced image then is filtered through a library of up to 1,024 binary pixel masks (each up to 25x25 in size) that define ROIs within fluorescence is summed to extract calcium traces;

Step 4: Decoding. Population vectors of extracted calcium trace values are stored in the DRAM, and then decoded by sending them as inputs to a linear classifier running on the MPSoC's ARM core.

Experimental methodology:

Step 1: collect an initial imaging dataset and store it on the host PC;

Step 2: pause for an intermission to identify cell contours if necessary (this is only required for contour-based decoding; see below) and train a linear classifier to decode behavior from the initial dataset;

Step 3: upload classifier weights from the host PC to the Ultra96;

Step 4: perform real-time decoding with the trained classifier.

---

## [Author Response]

Essential revisions:1) Provide a clear estimate of position accuracy that can be achieved with a space metric and not with a classifier accuracy with space bins whose size is unclear.

The size of the position bins (~20 cm) is now clarified in the main text (page 13) as well as in the caption for Figure 3A. As requested, position decoding accuracy is now reported using a space metric (cm) in addition to the classifier hit rate (Figures 3E-F, Figure 4A). To accommodate this change, it was necessary to alter the decoder’s output representation from the ordinal scheme used in the original submission (which was discontinuous at the ends of the track) to a Gray code in the revised submission (which has no discontinuities and thus allows distance error to be measured for all position bins, including at the ends of the track). The Gray coding scheme is illustrated in Figure 3B and described on page 13 the main text.

Compare position estimates to what would be obtained from electrophysiology based on the literature.

To address this, we have used a publicly available dataset of single-unit spike trains from place cells (n=53) recorded on a linear track; the data is from Hector Penagos in Matt Wilson's lab at MIT (see ref #30 in the revised manuscript). Figure 7E shows that for a subset of our CA1 imaging rats (n=3) in which the number of detected CB (contour-based) traces was close to 50 (and thus similar to the number of single units in the Penagos/Wilson place cell dataset), the single unit spike decoder’s mean distance error is 15-25 cm better than the calcium decoder trained on ~50 CB traces, and 5-10 cm worse than the than the calcium decoder trained on 900 CF (contour-free) traces. These results show that the real-time calcium decoder is able to achieve distance errors within a similar range as those achievable with offline analysis of place cell spike trains.

2) Please provide a spatial distribution of speed, acceleration, arrests, and a description of the number of place cells and of the size of the place fields.

Spatial distributions of speed, acceleration, and arrests (time spent sitting still) are now plotted in panel C of the Supplement to Figure 3.

As for the “number of place cells,” we did not classify calcium traces as originating from place versus non-place cells (indeed, CF calcium traces can combine fluorescence from multiple neurons, and thus cannot be sourced to single cells at all). Instead, we decoded the rats’ position from all available calcium traces (that is, all ROIs) of a given type (Off, CB, CF, CB+) regardless of their spatial tuning properties. This was not adequately explained in the original manuscript, so we have made it more clear in the revised paper by explaining this on pages 12 and 14., and by revising language about “place cells” throughout.

As in the original paper, the revised paper reports how many calcium traces of each type were extracted during all imaging sessions in all rats (Figure 3E, Figure 5B, bottom). Again, we do not count or report the number of “place cells” since we did not define a decision boundary for classifying whether or not a given calcium trace originated from a “place cell.”

To compute the “size of the place fields” as requested by the referees, despite not classifying calcium traces into place versus non place cells, we derived a ‘calcium trace activity range’ (CTAR, defined as total distance over which tuning curve exceeds ⅔ of its peak value) measure for each calcium trace in the dataset (Panel E in Supplement to Figure 3). These distributions show that CF traces were active over a significantly larger region of the track than other trace types, consistent with the fact that CF traces could combine fluorescence from more than one CA1 cell.

Related to that, it is crucial to provide a plot of the accuracy of spatial decoding against spatial location to appreciate the performance of the system in real conditions, knowing that in linear tracks place fields are not homogeneously distributed.

We agree that this is very important, which is why we included such plots in both the original and revised manuscripts. In the revised manuscript, mean decoding accuracy in each of the 24 position bins is shown in the polar plots on the left side of Figure F. Separate plots are shown for decoders trained on offline, CB, CF, and CB+ traces.

3) Improve the description of the methodology and of imaging speed with respect to displacement speed according to referee 2's comments. It is particularly important to explain how much exploration time is needed to train the real-time classifier.

Concerning “imaging speed with respect to displacement speed” see reply to comment #7 below. Concerning “how much exploration time is needed to train the real time classifier” Figure 3G of the revised manuscript shows how position decoding accuracy improves as a function of the size of the training dataset; it can be seen that decoding accuracy asymptotes when the training set reaches about 300 frames in size. Additional discussion of this issue is now provided on page 5 of the main text in the paragraph on ‘Step 2’ of the real-time experiment.

4) Provide an evaluation of the localisation information available in corrected motion artefacts in order to rule out that residual motion artefacts do not provide spatial information.

Regardless of whether analysis is done offline or online, any calcium imaging and decoding experiment is vulnerable to two potential problems arising from motion artifact:

Problem #1. Image motion can generate noise in calcium signals that disrupts the accuracy of decoding.

Problem #2. Image motion that is correlated with behavior can convey uncontrolled information that allows the decoder to learn predictions from image motion rather than calcium signals.

Very few published in-vivo calcium imaging experiments provide adequate controls for these two possible sources of artifact (again, such controls are just as necessary for offline as for online experiments). In response to the referee comments, we have provided controls for these confounds in our performance tests of DeCalciOn’s online decoding capabilities.

Figure 4B of the revised paper shows that without online motion correction, several rats in the linear track experiment show a significant correlation between position error and motion artifact (indicated by positive values on the y-axis); hence, motion artifact impairs decoding of position on the linear track in these rats (problem #1 above). This correlation between motion artifact and decoding error is reduced or eliminated by online motion correction (as indicated by values near zero on the x-axis), demonstrating that online motion correction helps to prevent motion artifact from impairing the accuracy of decoding.

Figure 6 of the revised paper shows that during an operant touchscreen experiment, motion artifact occurs preferentially during specific behaviors such as visiting the food magazine (reward retrieval, Figure 6A) or touching the screen to make a response (correct choice, Figure 6B). When motion correction is not used (top graphs in Figures 6C-F), the average motion artifact is higher during frames when the decoder accurately predicts behavior than during frames when the decoder fails to predict behavior; hence, motion artifact appears to improve the accuracy of predicting these behaviors (problem #2 above). When motion correction is used, the average motion artifact no longer differs for correctly versus incorrectly decoded frames (except in one case, bottom right graph of Figure 6E), indicating that motion correction helps to prevent the decoder from learning to predict behavior from motion artifact.

5) Repeat the experiment in a more complex environment, such as a 2D arena, in order to demonstrate the versatility of the method.

We agree that versatility is important, because DeCalciOn is intended for use by a wide range of experimenters studying neural activity during various kinds of behaviors. That being the case, a shortcoming of the original manuscript was that it only reported performance testing for position decoding from CA1 neurons and no other behaviors, perhaps conveying the false impression that the system is specialized for position decoding.

Adding yet more position decoding examples (e.g., in a 2D environment as some referees requested) might only serve to reinforce inaccurate impressions that our system is specialized for position decoding. So rather than adding more examples of position decoding, we have diversified our performance testing by presenting results for decoding calcium activity from a different brain region (OFC rather than CA1) during a different kind of behavior (an instrumental touchscreen task rather than a linear track). These performance tests are reported in a new subsection of the Results titled “Decoding instrumental behavior from orbitofrontal cortex neurons” (pages 21-25) and in new figures (Figures 5,6).

6) Provide a direct comparison of real performances based DeCalciOn to two previous online calcium imaging algorithms. The technical innovations of this work would be better highlighted by directly testing all three of these algorithms, ideally on similar datasets.

We understand the logic of this request, but it is unfeasible for several reasons. One of the two cited online algorithms (ref #14) is supported by public resources on Github (similar to the public resources we have made available for our own system), but as noted in our revised Discussion section (pages 33-34), that system is designed for compatibility with benchtop 2P microscopes so we would have to engineer a solution for interfacing it with miniscopes to carry out the requested performance comparisons; we lack the time and resources for such an enterprise. The other cited online algorithm (ref #25) is supposed to be compatible with UCLA Miniscopes, but public resources for this algorithm are not available. Other cited online algorithms (refs #6,#11,#26) were not performance tested in hardware at all.

We very much appreciate and respect all of the referee’s critiques, and have made diligent efforts to address most of them in revision. We will respect the final decision to accept or reject our revised paper on its scientific merits. We ask that as the referees and editors consider their decision, please take into account our view that comment #6 asks us to meet standards not met by prior cited publications (namely, direct hardware performance comparisons with other systems that either have never been implemented in hardware, are incompatible with our hardware, or are not publicly available).

7) Provide a quantification of the spatial error (across different locations, and running speeds) made in a virtual experiment in which the goal is to emit a TTL signal at a specific location of the animal. The difference with point 7 is that this quantification includes not only classifier inaccuracies but also the time delay of the real time device in relation to the speed of the animal.

We have addressed this by providing a much more thorough accounting of the latency between detection of photons at the miniscope sensor and generating of the TTL output pulse from the decoder (Figure 7 and subsection titled “Real time decoding latency” on pages 25-30 of the revised manuscript). As explained at the top of page 30:

“if we write x(t) to denote the position a rat occupies at the moment when a calcium trace vector arrives at the sensor, and x(t+tau_F) to denote the position the rat later occupies at the moment when TTL feedback is generated from the decoded calcium trace vector, the maximum distance error that can accrue between detection and decoding of the trace vector is δ_x=|x(t)-x(t+tau_F)| = 7.5 cm.”

Reviewer #1 (Recommendations for the authors):– Provide a clear estimate of position accuracy that can be achieved with a space metric and not with a classifier accuracy with space bins whose size is unclear.

This issue was addressed in response to essential revision #1a.

– Provide measurements in 2D and estimates of both location and head direction. How does it compare with electrophysiology? Are the conclusions about the lack of importance of cell body identification still holding for these conditions?

As noted in our reply to reviewer #1’s public comments, maximizing the system’s spatial position decoding capabilities was a lower priority design goal than rapidly disseminating a system with versatile capabilities for short latency decoding of various behaviors. To better demonstrate versatility, we have added a new section to the Results that shows categorical classification of behaviors during an instrumental touchscreen task (see above response to Essential Revision #5).

Reviewer #2 (Recommendations for the authors):– There is not enough information about the behavior. Please provide a spatial distribution of speed, acceleration, and arrests. Is there an influence on the decoding accuracy?

Distributions of speed, acceleration, and arrests are now provided in panel C of the Supplement to Figure 3.

– There is not enough information about the place cells. How many place cells? What is the spatial information of the cells? What are the size of the place fields?

Distributions of spatial stability scores and calcium trace activity ranges (CTAR) are now provided in panels D,E of the Supplement to Figure 3.

– Some details are lacking or are not explicitly described. For instance, I think the following information should be easier to obtain in the article. The 2.5m linear track is split into 13 regions, leading to regions of roughly 20cm.

This information is now clearly stated in the main text and in the caption for Figure 3A.

In linear tracks rats run with an average speed of 70cm.s^-1^, with maximum speed going up to 100cm.s^-1^. This means that running rats will stay between 280ms to 400ms in each zone. The video frames are down-sampled to 15 frames/s leading to 66ms between each frame. This implies that 4 to 6 frames will correspond to the same location.

The original methods section erroneously stated that video frames were downsampled to 15 frames/s, due to a copy-paste error from a description of offline analysis methods we have used in other experiments. We thank the referee for catching this error. It has now been corrected, and a detailed description of how far the rat can travel in a single frame (~7.5 cm) is provided in the main text at the top of page 29.

– A clearer description of the methodology might help the reader:Here is a short overview of the methodology:Technological methodology:Step 1: Correct for translational movement of brain tissue. Aa 128x128 pixel area with distinct anatomical features was selected within the 512x512 cropped subregion to serve as a motion stabilization window;Step 2: ACTEV removes background fluorescence from the 512x512 image;Step 3: ROI definition: The enhanced image then is filtered through a library of up to 1,024 binary pixel masks (each up to 25x25 in size) that define ROIs within fluorescence is summed to extract calcium traces;Step 4: Decoding. Population vectors of extracted calcium trace values are stored in the DRAM, and then decoded by sending them as inputs to a linear classifier running on the MPSoC's ARM core.Experimental methodology:Step 1: collect an initial imaging dataset and store it on the host PC;Step 2: pause for an intermission to identify cell contours if necessary (this is only required for contour-based decoding; see below) and train a linear classifier to decode behavior from the initial dataset;Step 3: upload classifier weights from the host PC to the Ultra96;Step 4: perform real-time decoding with the trained classifier.

We have followed the referee’s advice to structure the methodological description in this way, except the order of technological and experimental methodology reversed. The first subsection of the Results (titled ‘Steps of a real time imaging session’) and panel B of Figure 1 provide a detailed description of the experimental methodology (note that for additional clarity we now break the session down into 5 steps instead of 4), and the second subsection of the Results (titled ‘Real time image processing pipeline’) provides a detailed description of the technical methodology.